# Single-cell eQTL mapping of human endogenous retroviruses reveals cell type-specific genetic regulation in autoimmune diseases

Fan Zhu[1,2], Yi Liu[1,2], Jiehao Lei[1,2], Xinxing Li[1], Zexu Jiang[1], Xuan Dong [1,3], Ying Gu [1,3,4,5] & Young Li [1,3,6] ✉

Human endogenous retroviruses constitute a significant portion of the human genome and play complex roles in gene regulation and disease processes. However, the expression pattern and disease associations of specific retroviral loci remain poorly understood. This study examines the expression and regulatory mechanisms of these retroviral elements in immune cells. Utilizing single-cell RNA sequencing data from peripheral blood mononuclear cells, we identify 41,460 expressed retroviral loci, with 1936 showing cell type-specific expression. We further detect 3463 conditionally independent expression quantitative trait loci linked to retroviral elements, highlighting their potential role in mediating genetic variants and disease associations. Notably, these retroviral sequences associate significantly with autoimmune diseases, with specific loci demonstrating pleiotropic associations with disease-related genes. Our findings suggest that these elements are not merely genomic remnants but active participants in cellular regulation and disease progression. This work advances understanding of human-retroviral coevolution and highlights potential therapeutic targets in immune disorders.

Human endogenous retroviruses (HERVs) are remnants of ancient retroviral infections, constituting ~8% of the human genome with over 500,000 loci[1]. Structurally similar to exogenous retroviruses, intact HERV proviruses are characterized by the sequence: 5'LTR-gag-pro-pol-env-3'LTR[2]. However, due to accumulated mutations and deletions over evolutionary time, the majority of HERV elements in the human genome are incomplete, often lacking one or more of these essential genes[3]. Among the various forms of HERVs, solo LTRs (long terminal repeats) are the most prevalent, representing a significant fraction of these genomic elements. Additionally, host epigenetic repression further contributes to the inactivation of most HERV loci, rendering them transcriptionally silent under physiological conditions and generally considered harmless to humans[4]. Despite their widespread inactivation, some HERV loci have been co-opted by the host genome to serve indispensable roles. For instance, the env genes of HERV-W and HERV-FRD produce syncytin-1 and syncytin-2, respectively, which are crucial for the formation of placental syncytia and the maintenance of trophoblast cell fusion[5]. Studies have shown that LTRs can function as promoters or enhancers, regulating the expression of adjacent genes. For example, ERVs have been shown to act as enhancers of IFN (interferon) genes, thereby contributing to the evolution of transcriptional networks for the IFN response, a major component of innate immunity[6]. Despite these programmed incorporations of HERV elements in normal cellular processes, some HERV loci may become

[1]BGI Research, Hangzhou, China. [2]College of Life Sciences, University of Chinese Academy of Sciences, Beijing, China. [3]State Key Laboratory of Genome and Multi-omics Technologies, BGI Research, Hangzhou, China. [4]BGI Research, Shenzhen, China. [5]State Key Laboratory of Genome and Multi-omics Technologies, BGI Research, Shenzhen, China. [6]Zhejiang Key Laboratory of Spatial Omics, Hangzhou, Zhejiang, China. ✉e-mail: liyang13@genomics.cn

reactivated, which may contribute to tumorigenesis[7,8], autoimmunity[9,10], senescence[11], and aging[12]. Therefore, a comprehensive understanding of the expression and regulatory mechanisms of HERVs would significantly advance our knowledge of the roles HERVs play in humans.

Our immune system is essential for protecting us from exogenous viral infections, and studies have shown that HERVs can also interact with the immune system. For example, the reverse transcription products of HERVs may induce aging and inflammatory responses by activating the cGAS-STING innate immune pathway[11]. Additionally, HERV-derived RNAs can act as immunogenic molecules, triggering pattern recognition receptors (PRRs) such as Toll-like receptors (TLRs) and RIG-I-like receptors (RLRs), thereby eliciting antiviral immune responses[13]. Furthermore, HERVs can contribute to immune dysregulation by producing viral-like proteins or peptides that mimic self-antigens, potentially leading to autoimmune responses[14]. Researchers have also observed a higher HERV expression in the peripheral blood mononuclear cells (PBMCs)[15] of aged individuals, suggesting a potential link between HERV activity and immunosenescence. Despite these findings, the expression patterns, activation mechanisms, and specific roles of HERVs in immune cells remain largely unknown.

In this study, we developed a HERV analysis pipeline for single-cell RNA-seq data and analyzed the HERV expression patterns in 981 publicly available PBMC samples[16] (OneK1K, GSE196830). We identified 41,460 expressed HERV elements in PBMCs and observed cell type-specific expression of HERVs. Utilizing genotype data, we further investigated genetic variants that regulate HERV expression, identifying 2888 single-nucleotide polymorphisms (SNPs) and 1805 corresponding HERVs as expression quantitative trait loci (eQTLs), most of which were cell type-specific. Moreover, by integrating disease-associated genome-wide association study (GWAS) data, we found that dysregulated HERV expression in immune cells may be associated with various immune-related diseases, particularly autoimmune diseases. We further illustrated that one potential mechanism underlying the HERV-disease associations could be that HERVs may influence the expression of autoimmune disease-related genes. Overall, our results suggest that cell type-specific HERVs are important mediators of the genetic effects of immune-related diseases, especially in the context of autoimmune disorders.

## Results

### HERVs are expressed independently of genes in PBMCs

To investigate the expression patterns of HERVs in immune cells, we utilized the OneK1K single-cell RNA-seq dataset, which includes PBMCs from 981 healthy donors aged 20–90 years, encompassing a total of 1.2 million single cells. We implemented the following analytical pipeline: First, we obtained HERV annotations from the UCSC Table Browser (GRCh38/hg38 assembly) and merged them with GENCODE (v43) protein-coding gene annotations. To distinguish independent HERV transcription from host gene expression, we excluded all HERV loci overlapping with gene exons (Supplementary Data 1). This conservative filtering strategy ensured our analysis focused specifically on HERV elements with potential autonomous regulatory activity. Using CellRanger[17] (v7.1.0), we then constructed a combined reference incorporating both filtered HERV annotations and protein-coding genes through the "mkref" function. Finally, we quantified gene and HERV expression levels for each single cell using this combined reference with stringent unique mapping criteria. (Fig. 1a). Based on canonical gene expression patterns, we identified and annotated nine distinct immune cell populations: CD4-T cells, CD8-T cells, B cells, natural killer cells (NK), myeloid cells, plasma cells, γδ T cells (gdT), megakaryocytes (Mega), and hematopoietic stem cells (HSC) (see the "Methods" section; Fig. 1b). To address the high repetitiveness of HERVs, we configured CellRanger[17] (version 7.1.0) to retain only unique mapping reads (Fig. 1c). This step minimized the impact of repeat

counting and other artifacts associated with repetitive elements. We further assessed the proportion of unique reads for each HERV and found that most HERVs exhibited a proportion close to 100% (Supplementary Fig. 1a), indicating that HERV expression was predominantly driven by unique reads. For example, in a single cell, *Harlequin-int_dup64-chr1* and *Harlequin-int-chr17*, a total of 4 and 5 reads were initially mapped to these two HERVs, respectively. Among these, one read was mapped to both HERVs. After filtering out multi-mapping reads, only 3 and 4 uniquely mapped reads remained, demonstrating the effectiveness of our approach in handling repetitive elements (Fig. 1d). Our approach to quantifying HERV expression at the locus level overcomes key limitations of existing methods. Unlike scTE[18] (family-level aggregation) and soloTE[19] (random assignment of multi-mapping reads), which either lack locus-specific resolution or introduce bias, and CELLO-Seq[20], which is limited to small-scale datasets, our method retains only unique mapping reads, ensuring high accuracy and scalability (Supplementary Fig. 1b). This approach has been validated in similar studies[21], further supporting its robustness and applicability.

Due to the overall low expression frequency of HERVs, we applied a lenient threshold of >20 cells to retain as many HERVs as possible for downstream analysis (Supplementary Fig. 1c). This approach ensures that we do not miss potentially biologically relevant HERVs that may play a role in cell type-specific regulation or disease associations. After filtering out HERVs expressed in fewer than 20 cells, we identified 41,460 expressed HERV loci in PBMCs (Supplementary Data 2).

The length distribution of expressed HERV loci was similar to that of the entire genome, with a significant enrichment in LTR forms, as previously reported[22] (Fig. 2a). We examined the expressed HERV families and found that ERV1 and ERVK families were more active in PBMCs (Fig. 2b), possibly due to their relatively recent integration into the human genome[23]. To further investigate the expression patterns of HERV families across different cell types and age groups, we analyzed the proportion of each HERV family in individual cell types (Supplementary Fig. 2a) and their distribution across age groups within each cell type (Supplementary Fig. 2b). Notably, our analysis revealed no significant differences in HERV family proportions across age groups within any cell type, suggesting that the expression of HERV families is relatively stable over time within specific immune cell populations. We further annotated the genomic regions of the expressed HERVs, revealing that most were located in intronic regions, with 9713 (23%) on the same strand and 10,591 (26%) on the opposite strand of related genes (Fig. 2c). Additionally, 18,324 (44%) expressed HERVs were found in intergenic regions (Fig. 2c). For example, *MLT1F1_dup14_chr9*, situated in an intergenic region near *KLHL9*, exhibited a prominent read peak on the RNA expression track (Fig. 2d), suggesting programmed expression rather than sequencing noise.

To mitigate the impact of sequencing depth (Supplementary Fig. 3a), we normalized HERV expression against total counts (see the "Methods" section, Supplementary Fig. 3b, c). We then assessed whether gene expression influences HERV expression by calculating the Spearman's correlation between each expressed HERV and its nearest genes. Our analysis revealed that most HERVs exhibited low correlation with adjacent genes (Fig. 2e). For instance, *IDNK* and *VMAC* were consistently expressed in the CD4-T, CD8-T, NK, B and myeloid cells, whereas the expression of *LTR101_Mam_dup28-chr9* and *THE1C_dup7-chr19* located in introns, varied among these different cell types (Fig. 2f, Supplementary Fig. 3d). These findings demonstrate that HERVs are expressed independently from genes in PBMCs.

### Cell type-specific epigenetic states shape different HERV profiles

We observed that HERV expression exhibits significant variation across different cell types (Fig. 2f, Supplementary Fig. 3d). To further investigate, we compared HERV expression profiles between cell types and

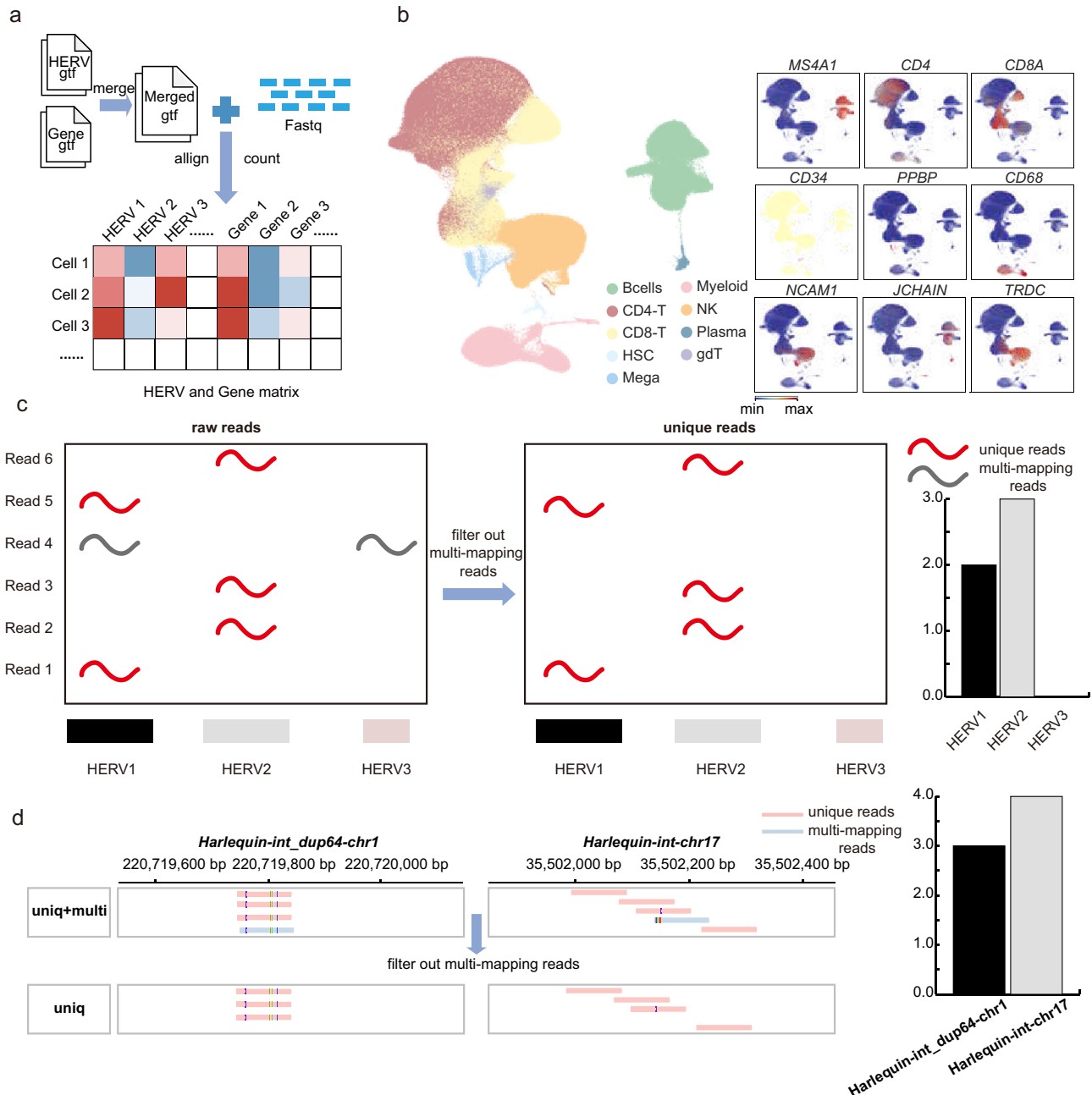

**Fig. 1 | Single-cell RNA-seq analysis and HERV quantification in PBMCs. a** Flow chart of HERV and Gene quantification. **b** Single-cell RNA-seq UMAP projection of 1.2 million PBMCs across all individuals, showing 9 transcriptionally distinct populations, based on gene expressions. Colors are coded to indicate the cell types. **c** Strategies for quantifying HERVs. **d** Example of read filtering in a single cell: For *Harlequin-int_dup64-chr1* and *Harlequin-int-chr17*, a total of 4 and 5 reads were initially mapped, respectively. Among these, one read was mapped to both HERVs. After filtering out multi-mapping reads, only 3 and 4 uniquely mapped reads remained.

found that HERV profiles showed markedly lower similarity compared to gene expression profiles across these cell types (Fig. 3a). Based on the dispersion and mean expression of HERVs, we identified 2045 highly variable HERVs, which showed low correlation with adjacent genes (Fig. 3b). UMAP dimensionality reduction based on these highly variable HERVs revealed distinct spatial segregation of CD4-T, CD8-T, NK, B, and myeloid cell populations in the low-dimensional embedding (Fig. 3c). Subsequent differential expression analysis revealed 1936 cell type-specific HERVs (Fig. 3d, Supplementary Data 3). For instance, *LTR7_dup7-chr12* was predominantly expressed in NK cells and minimally in CD8-T cells, while *LTR33_dup233-chr20* and *MLT1H2_dup140-chr2* were mainly expressed in CD4-T and B cells, respectively (Fig. 3e).

As shown in Supplementary Fig. 1c, while the majority of HERVs are detected in a relatively small number of cells (typically 20–100 cells), the cell type-specific HERVs reported in our manuscript are expressed in a significantly higher number of cells, often ranging from 1000 to 10,000 cells. This indicates that the cell type-specific effects we describe are driven by HERVs expressed at higher frequencies within specific cell populations, rather than low-frequency artifacts.

We further annotated these cell type-specific HERVs using the PBMC chromatin state data downloaded from Roadmap Epigenomics Project (https://egg2.wustl.edu/roadmap/web_portal/chr_state_learning.html#core_15state)[24]. The analysis revealed that cell type-specific HERVs expressed in immune cells were primarily located in

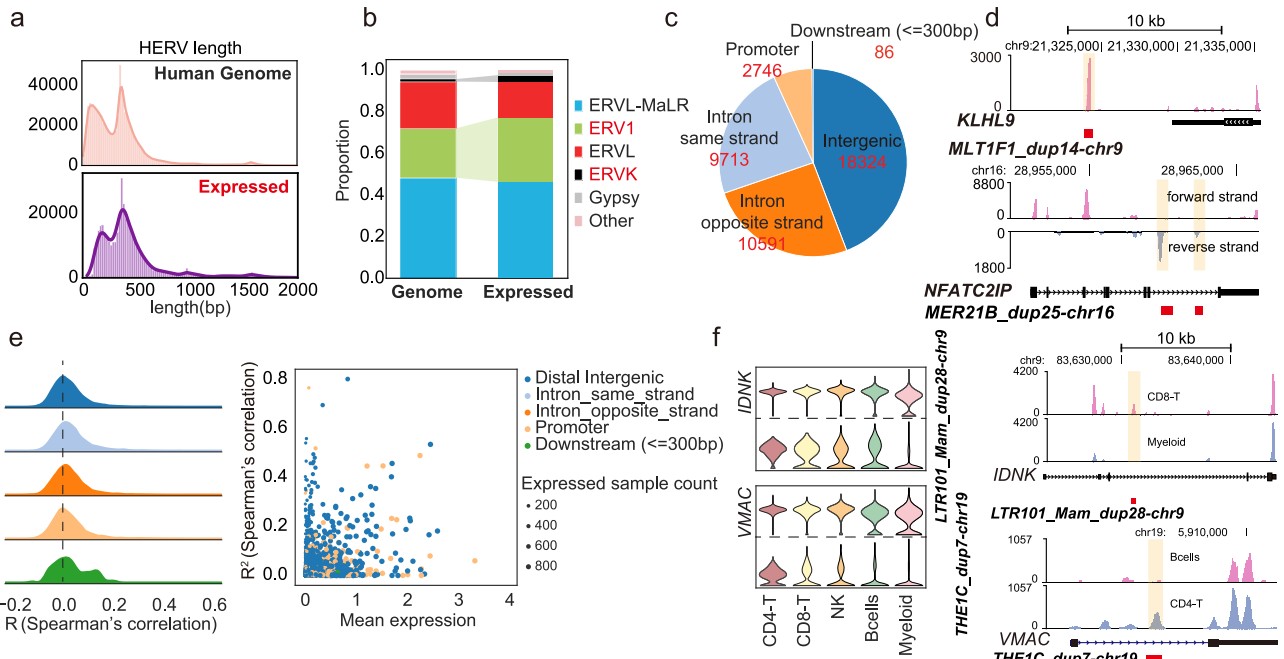

**Fig. 2 | Comprehensive characterization of HERVs. a** Length distribution of HERVs. **b** Distribution of HERVs across different families. **c** Proportion of genomic regions where HERVs are located. **d** Distribution of RNA-seq peaks in the *KLHL9* and *NFATC2IP* locus. *MLT1F1_dup14-chr9* is located in the intergenic region, and *MER21B_dup25-chr16* is located in the antisense strand of the *NFATC2IP* intron. **e** Spearman's correlation coefficient distribution of HERVs expression and nearby genes. Most HERVs show low correlation with related genes. **f** Left: expression levels of *IDNK*, *VMAC* and HERVs in five cell types. Right: distribution of RNA-seq peaks in the *IDNK* and *VMAC* loci.

active chromatin regions (60–80%) (Fig. 3f). Enrichment analysis confirmed that these cell type-specific HERVs were markedly enriched in active chromatin regions (Fig. 3g). Additionally, we examined cell type-specific enhancers and promoters annotated by H3K27ac and H3K4me3 ChIP-seq data from the ENCODE database[25], finding that cell type-specific HERVs were highly enriched in corresponding cell type-specific enhancers or promoters (Fig. 3h). For example, HERV *MLT1D_dup92-chr1* is located on a CD8-T cell enhancer and is predominantly expressed in CD8-T cells, while NK-specific HERV *PRIMA4_LTR_dup3-chr14* is situated in the promoter region of NK cells (Fig. 3i).

To validate the enrichment of cell type-specific HERVs in active chromatin regions, we performed independent validation analyses using another single-cell RNA-seq dataset of PBMCs (syn50209110)[26]. Consistently, we observed cell type-specific HERV profiles that were highly consistent between the two datasets (Supplementary Fig. 4a–c). The cell type-specific HERVs showed consistent enrichment in the active chromatin regions of corresponding cell types (Supplementary Fig. 4d, e). These findings indicate that immune cells possess distinct HERV expression profiles, shaped by their unique epigenetic states. This provides evidence that HERVs can function as regulatory elements, such as promoters and enhancers[27].

**Epigenetic states determine the cell type-specific HERVs *cis*-eQTLs**
In the original study of the OneK1K cohort, the authors demonstrated that genetic variations can influence gene expression in a cell type-dependent manner. Building on this, we hypothesized that HERV expression might also exhibit cell-type-specific expression quantitative trait loci (eQTLs). We employed TensorQTL[28] to examine the association between HERV expression and SNPs within a 1 Mb *cis*-region upstream and downstream of the HERVs across different cell types. Our analysis identified a total of 3463 conditionally independent *cis*-eQTLs across five immune cell types (see the "Methods" section, Fig. 4a, Supplementary Data 4), involving 1805 HERVs (hereafter termed as eHERVs) and 2888 SNPs (hereafter termed as eSNPs)

(Supplementary Fig. 5a, b). Consistent with the original OneK1K study, we observed a bias in the number of eHERVs across cell types, with 1652 identified in CD4-T cells, compared to only 157 in myeloid cells (Supplementary Fig. 5a). This discrepancy is primarily due to the varying cell numbers across different cell types, which directly impacted the statistical power of eQTL tests[29] (Supplementary Fig. 5c).

Our study substantially advances the HERV eQTL analysis performed by She et al.[30] in several key aspects. First, our analysis is based on single-cell RNA sequencing data from PBMCs of 981 healthy donors, compared to 315 blood samples in She et al. This larger sample size enabled us to identify 1805 eHERVs, a substantial increase over the 263 eHERVs reported in their study. Second, while She et al. identified *cis*-eQTLs at the bulk tissue level, our study provides cell type-specific *cis*-eQTLs, providing higher resolution insights into HERV regulation across different immune cell types. These advancements highlight the added value of our work in uncovering the regulatory landscape of HERVs at a higher resolution and with greater statistical power.

Our findings indicate that HERV eQTLs are predominantly cell type-specific, with 2736 out of 3463 (79%) specific to individual cell types, and the majority identified in CD4-T cells (Fig. 4b, c). For instance, *THE1D_dup769-chr3* shows a significant eQTL effect with rs6780016 exclusively in CD4-T cells (*P*-value = 9.8e−7), whereas the eQTL effect between *LTR9_dup12-chr6* and rs28894981 is detected only in NK cells (*P*-value = 1.9e−6, Fig. 4d). We further investigated whether these cell type-specific eHERVs exhibited cell type-specific expression. Interestingly, we found that most cell type-specific eHERVs were expressed in multiple cell types (Supplementary Fig. 5d, e), suggesting that the cell type-specific *cis*-eQTL effects on HERVs expression are not due to differential expression of HERVs.

To understand how eSNPs regulate eHERVs expression, we examined the relative distance of eSNPs loci to their corresponding eHERVs. Despite setting our *cis*-regulation genome range to 1 Mb, we found that the majority of eSNPs were located within ~200 kb of the transcription start sites (TSSs) of target eHERVs (Fig. 4e). Further annotation of the chromatin states of these eSNPs loci revealed a high

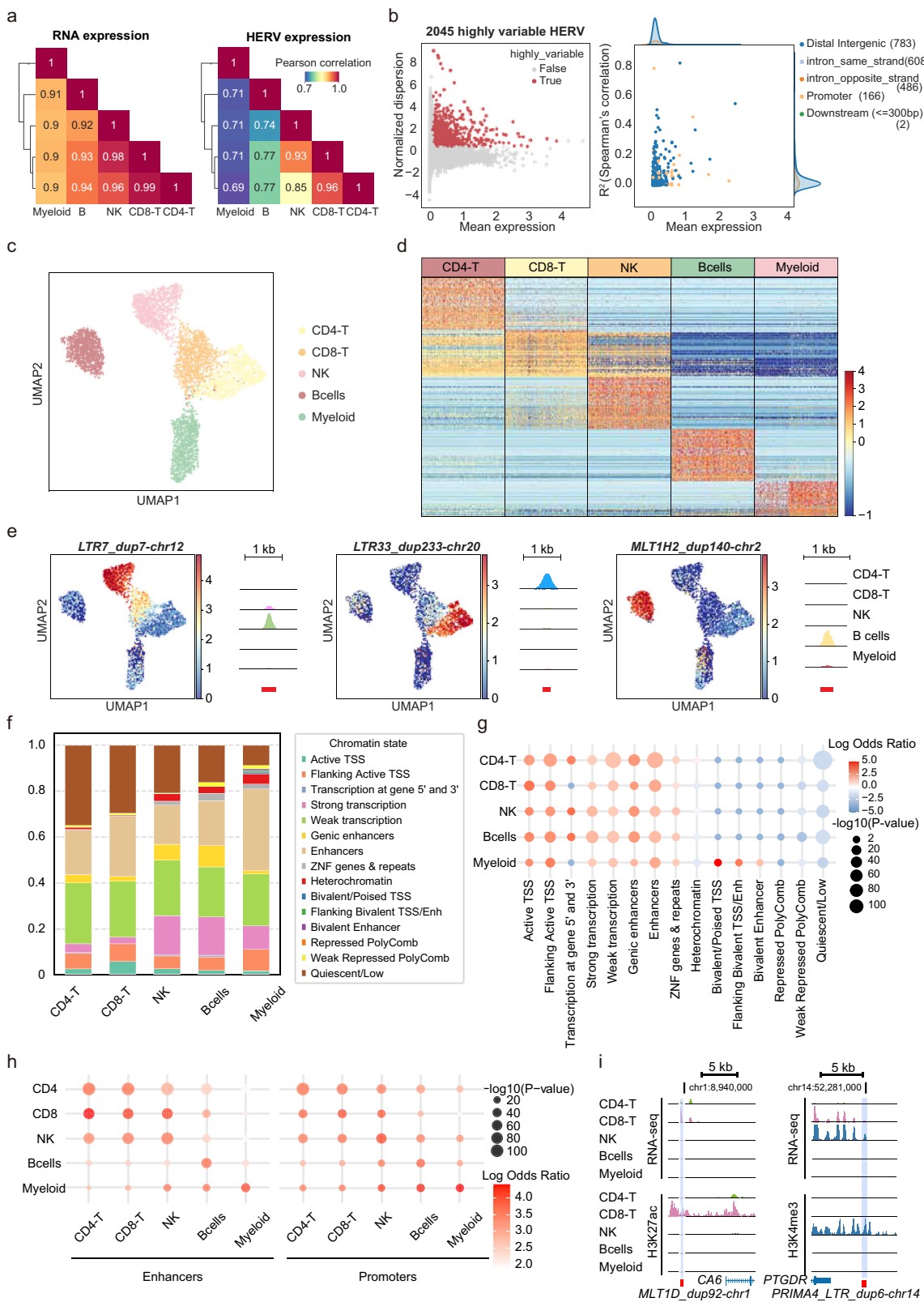

enrichment in active chromatin regions (Supplementary Fig. 5g). Notably, we observed a greater enrichment of eSNPs in cell type-specific enhancers (Fig. 4f). For example, eSNP rs6780016 resides in a CD4-T cell-specific enhancer region, 53 kb downstream of the target eHERV *THE1D_dup769-chr3*. Although *THE1D_dup769-chr3* is expressed in all five cell types, a significant eQTL effect between rs6780016 and *THE1D_dup769-chr3* was only detected in CD4-T cells (Fig. 4g). These

results suggest that cell-type-specific HERV *cis*-eQTLs are likely driven by the distinct epigenetic states of different cell types.

## HERVs are highly associated with autoimmune diseases

Both genetic variants and abnormal HERV expressions have been reported to play critical roles in various conditions, including auto-immune diseases[9,10,31], cancers[7,8,32], and aging[11,12]. However, the

**Fig. 3 | Characterization and genetic states of cell type-specific HERV expression. a** Expression similarity of Genes and HERVs in five cell types. Left: Gene, Right: HERV. **b** Left panel: scatter plot of highly variable HERVs (dispersion > 0.5, expression > 0.1). Right panel: scatter plot of the Spearman's correlation coefficient of highly variable HERVs with related genes. **c** UMAP projection based on the expression of highly variable HERVs. Each spot represents a pseudobulk cell type sample of an individual. **d** Heatmap of cell type-specific HERV expression. **e** UMAP projection (left) and RNA-seq track (right) of cell type-specific HERV expression. **f** Proportion of 15 chromatin states of cell type-specific HERVs across immune cells. **g** Enrichment of cell type-specific HERVs across 15 chromatin states corresponding to each immune cell type. *P*-value and Odds ratio were calculated using two-sided

Fisher's exact tests comparing the observed overlap of HERVs with each chromatin state against background expectations. Multiple testing correction was performed using the Benjamini−Hochberg false discovery rate (FDR) method (threshold = 0.05) implemented via the p.adjust function in R. **h** Enrichment of cell type-specific HERVs within the promoter and enhancer regions of each corresponding cell type. *P*-value and Odds ratio were calculated using two-sided Fisher's exact tests comparing the observed overlap of HERVs with each promoter and enhancer region against background expectations. Multiple testing correction was performed using the Benjamini−Hochberg false discovery rate (FDR) method (threshold = 0.05) implemented via the p.adjust function in R. **i** RNA-seq, H3K27ac, H3K4me3 Chip-Seq track of *MLT1D_dup92-chr1* and *PRIMA4_LTR_dup3-chr14* in each cell type.

potential synergistic effects of these factors in promoting disease occurrence and progression remain unclear. To explore this, we examined the enrichment of HERV-associated eSNPs identified from PBMCs in disease-associated SNPs from the GWAS catalog[33] (https://www.ebi.ac.uk/gwas/). As expected, we found that these HERV-associated eSNPs show significant enrichment in immune-related diseases, such as autoimmune diseases and cancers (Supplementary Fig. 6a). To further investigate, we collected GWAS data for 81 immune-related diseases, including 21 autoimmune diseases, 24 cancers, 11 inflammatory diseases, 6 allergic diseases, and 19 infectious diseases (Supplementary Data 5). We then employed summary-data-based Mendelian randomization (SMR) analysis to assess the pleiotropic associations between eHERV-associated eQTLs and these immune-related diseases (Fig. 5a). Our analysis identified a total of 548 significant pleiotropic associations across five immune cell types (*q*-value < 0.05 and pHEIDI > 0.01), involving 196 eHERVs and 35 diseases (Supplementary Data 6). Notably, 76.2% of the autoimmune diseases (16 out of 21) were associated with 100 cell type-specific HERVs, with the highest proportion and association count among immune-related diseases (Fig. 5b). This suggests that HERVs may play a crucial role in the pathogenesis of autoimmune diseases.

Focusing on the association of 100 eHERVs with autoimmune diseases (Fig. 5c), we found that HERVs showed the strongest associations with autoimmune diseases of the digestive system, such as type I diabetes (T1D), celiac disease (CEL), Crohn's disease (CD), and ulcerative colitis (UC) (Fig. 5c). This association may be attributed to the digestive system's frequent exposure to exogenous viruses[34]. We also observed that some eHERVs showed significant associations with multiple autoimmune diseases. For instance, *MSTD_dup83-chr6* is associated with four autoimmune diseases: CEL, psoriasis (PSO), Sjogren's syndrome (SS), and autoimmune hepatitis (AH) (Fig. 5d). GeneHancer[35] records its interaction with 7 HLA genes (Supplementary Fig. 6b), consistent with the essential role of HLA genes in autoimmune diseases[36]. Interestingly, although B cells contained fewer cells and fewer eHERVs than NK cells (Supplementary Fig. 5c), they exhibited a higher proportion of autoimmune disease-associated HERVs (Fig. 5c). This discrepancy suggests potential specialized roles of HERVs in modulating B cell functionality and their contribution to autoinflammatory pathways."

Among these HERV-associated autoimmune diseases, CD has been reported to exhibit abnormal HERV expressions[37], but the specific responsible HERV loci and cell types remain unclear. We identified 20 HERVs associated with CD across CD4/CD8-T, B, NK, and myeloid cells (Supplementary Fig. 6c). For example, we found that *LTR2B_dup15-chr6* was negatively correlated with CD only in CD4-T cells, with increased expression of *LTR2B_dup15-chr6* associated with reduced susceptibility to CD ($\beta = -0.81$) (Fig. 5e). To further verify this association, we conducted genetic colocalization analysis, which tests whether two potentially related phenotypes share common genetic causal variants in a given genomic region. Our analysis revealed that *LTR2B_dup15-chr6* and CD were genetically associated at the causal SNP rs6939196 (SNP.H4 = 0.87), suggesting that *LTR2B_dup15-chr6* likely contributes to the pathogenesis of CD (Fig. 5f). We further compared the expression of *LTR2B_dup15-chr6* in single-cell RNA-seq data from

CD patients and healthy donors[38] (GSE157477), which showed that *LTR2B_dup15-chr6* was significantly expressed in CD4-T cells of CD patients, but not in other cell types (Fig. 5g, Supplementary Fig. 7a−c), indicating that *LTR2B_dup15-chr6* might be a CD4-T cell type-specific genetic causal effect of CD. These findings reveal that HERVs contribute significantly to autoimmune diseases.

## HERVs potentially regulate autoimmune disease-associated genes

As previously mentioned, HERVs can function as enhancers, alternative promoters, and other regulatory elements to regulate nearby gene expression. We hypothesized that disease-associated HERVs might contribute to disease progression by regulating disease-associated genes. To test this hypothesis, we conducted SMR analysis on autoimmune disease-associated HERV eQTLs and public gene eQTL results from the European population[39] to identify cell type-specific pleiotropic associations between HERVs and gene expression. Our analysis identified 168 potential HERV-gene regulatory interactions showing significant SMR associations, involving 104 HERVs and 70 genes across five immune cell types (Supplementary Fig. 8a, Supplementary Data 7). We found that these eHERVs might regulate gene expression in specific cell types, with the highest number (88) of significant HERV-gene interactions observed in CD4-T cells (Supplementary Fig. 8a, b). Additionally, we noted that a single HERV locus could be linked to the regulation of multiple genes, ranging from 1 to 3 (Supplementary Fig. 8b). For instance, *LTR12C_dup55-chr5* is associated with the regulation of *PPWD1* and *CENPK* exclusively in CD4-T cells (Supplementary Fig. 8c), while *MER4D1_dup36-chr5* is linked to the regulation of *TBCA* in both CD4-T and CD8-T cells (Supplementary Fig. 8d).

We further performed SMR analysis on these genes regulated by HERVs and autoimmune diseases (Supplementary Data 8). By combining the SMR results of HERV-gene, HERV-disease and gene-disease associations, we identified HERV-regulated genes that were also significantly associated with specific HERV-related diseases (Fig. 6a), involving 9 autoimmune diseases, 4 cell types, 8 HERVs and 4 genes (Fig. 6b). This suggests that the association of these genes with specific diseases may be mediated by the corresponding HERVs. For example, we identified *LTR2B_dup15-chr6* as a regulator of *RNASET2* in CD4-T cells (Fig. 6c), and SMR analysis showed that both *LTR2B_dup15-chr6* (Fig. 5e) and *RNASET2* (Fig. 6d) are genetically associated with CD. *RNASET2* encodes endoribonuclease RNaseT2, which plays a key role in combating exogenous pathogens[40]. Studies have revealed that polymorphisms in *RNASET2* are closely linked to CD severity[41] and may serve as diagnostic and therapeutic targets for CD[42]. Our data imply that HERV *LTR2B_dup15-chr6* likely mediates the link between *RNASET2* and CD.

Next, we explored the epigenetic regulation of *LTR2B_dup15-chr6* and *RNASET2*. We integrated H3K27ac, H3K4me3 ChIP-seq, and DNase-seq data of CD4-T cells from ENCODE[25]. We found significant enhancer signals at the *LTR2B_dup15-chr6* locus, located 1 kb upstream of *RNASET2* (Fig. 6e). Moreover, the GeneHancer[35] documents interactions between this enhancer and *RNASET2* (Fig. 6e). These data suggest that *LTR2B_dup15-chr6* plays a critical role in regulating the expression of

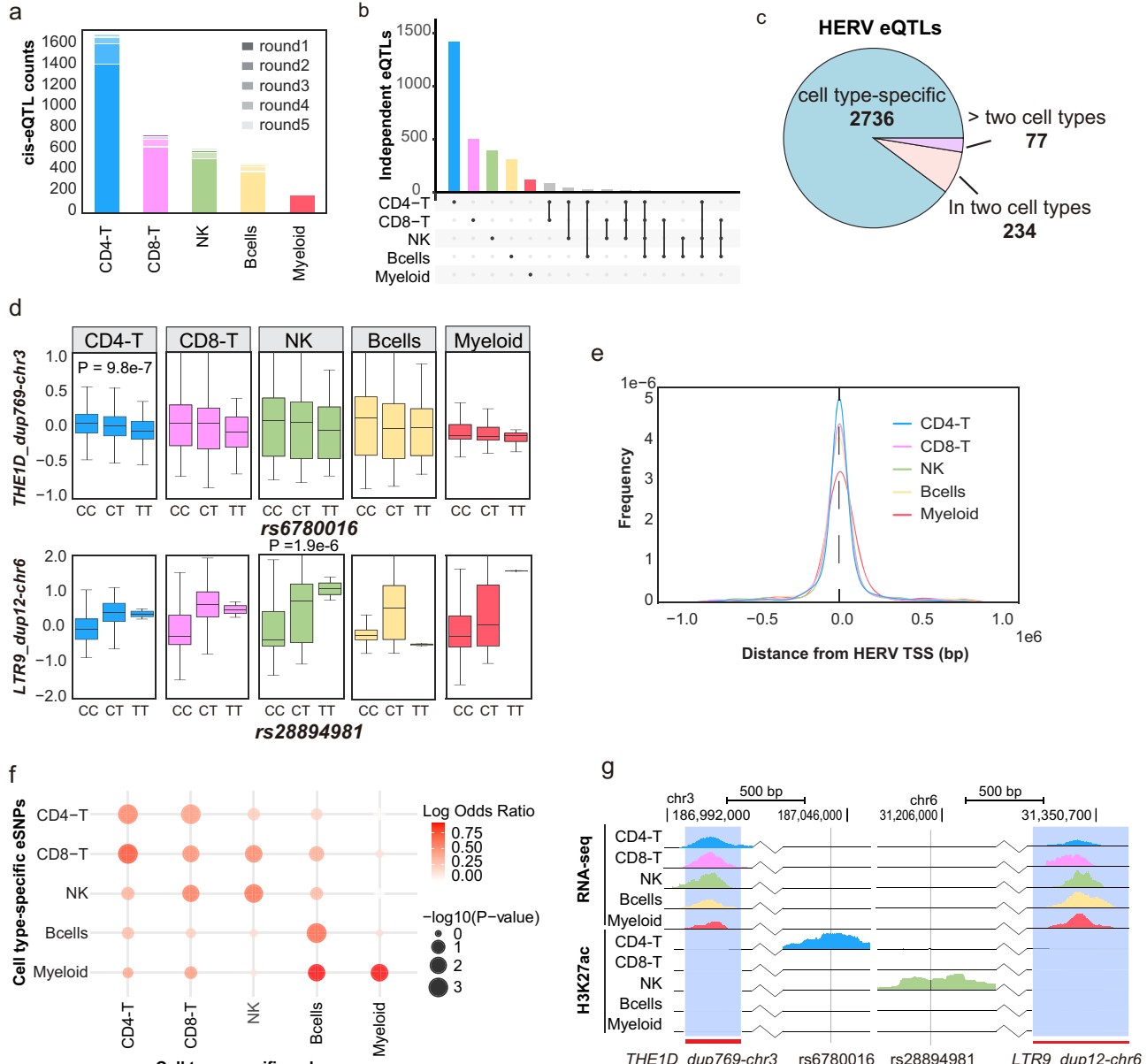

**Fig. 4 | Cell type-specific eQTLs of HERVs and their genetic states in immune cells. a** Bar plot showing the number of identified independent eQTLs of HERVs in each cell type. **b** Upset plot of unique and shared HERV eQTLs across different cell types. **c** Pie chart showing the proportions of unique and shared eQTLs. **d** Box plot showing the differential expression of cell-type-specific HERV eQTL examples across different cell types. Upper: CD4-T-specific eQTL between rs6780016 and *THE1D_dup769-chr3* expression. Lower: NK cell-specific eQTL between rs28894981 and *LTR9_dup12-chr6* expression. Statistical significance was determined using TensorQTL to evaluate allele dosage effects on cell-type-level HERV expression. *Cis*-regulatory associations were mapped through 10,000 permutations using the map_cis() function, generating empirical *p*-values and genome-wide false discovery rates. Center lines represent medians, box limits indicate the

25th and 75th percentiles, whiskers extend to 1.5× interquartile range (IQR) from the box edges. rs6780016 *n*(CC) = 539, *n*(CT) = 375, *n*(TT) = 67; rs28894981 *n*(CC) = 880, *n*(CT) = 99, *n*(TT) = 2. **e** Relative frequency distribution of the distance between eSNPs and the TSS of associated HERVs. **f** Dot plot showing the enrichment of cell type-specific eSNPs in cell type-specific enhancers. *P*-value and Odds ratio were calculated using two-sided Fisher's exact tests comparing the observed overlap of HERVs with each promoter and enhancer region against background expectations. Multiple testing correction was performed using the Benjamini–Hochberg false discovery rate (FDR) method (threshold = 0.05) implemented via the p.adjust function in R. **g** Genome track of RNA-seq and H3K27ac Chip-seq data around HERV *THE1D_dup769-chr3* and *LTR9_dup12-chr6*, and eSNP rs6780016 and rs28894981 in each cell type.

*RNASET2*. We further examined the expression patterns of *LTR2B_dup15-chr6* and *RNASET2* in single-cell RNA-seq data from CD patients and healthy donors[38] (GSE157477). As expected, the expression of *RNASET2* was significantly correlated with the expression of *LTR2B_dup15-chr6* (Fig. 6f), and both were downregulated in CD patients, specifically in CD4-T cells (Fig. 6g, Supplementary Fig. 8e). In conclusion, our results reveal the potential regulatory role of HERVs in disease-associated genes, which could be a crucial factor in disease progression, particularly in autoimmune diseases (Fig. 6h).

## Discussion

Studies have shown that HERVs are involved in various biological processes and complex diseases. These findings largely rely on observed differences in HERV family expression. In our study, we constructed a HERV-specific GTF file for mapping and identified expressed HERVs in PBMCs with locus-specific resolution. This allowed us to identify specific HERV loci associated with eQTLs, genes, and diseases. Notably, we excluded multi-mapped reads of HERVs due to the limitations of short-read sequencing in handling genomic

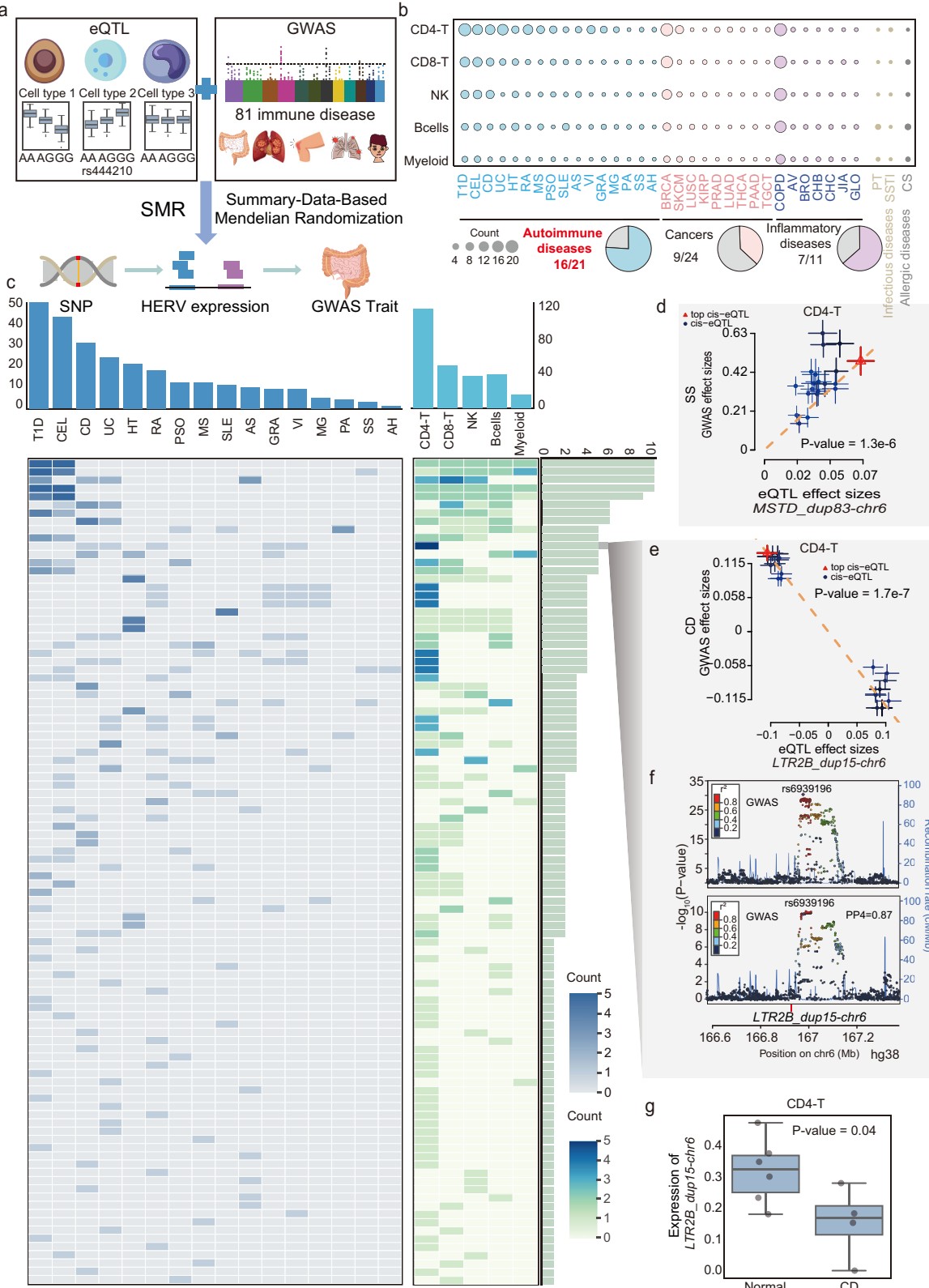

repetitiveness. Additionally, some HERV RNAs may lack polyA tails, potentially leading to their omission in polyT-based single-cell sequencing. Addressing these challenges requires technological innovations, such as the application of long-read sequencing and random RNA capturing in single-cell sequencing. Nevertheless, this locus-specific approach, building on established methodologies[21], could be extended to studies of other cell types, significantly

advancing our understanding of the functions of HERV elements in the human genome.

HERVs are categorized as endogenous retroelements within the human genome, reflecting their origin from ancient retroviral infections and their lack of active transposition. Recent studies have shown that Alu sequences, another class of endogenous retroelements, can embed in enhancer RNA (eRNA) and promoter upstream antisense RNAs (uaRNA)

**Fig. 5 | Pleiotropic associations of HERVs in immune-related diseases.**
**a** Schematic flow of our SMR analysis. The materials in the pictures were downloaded from "freepik.com", "Vectorportal.com", and "Vecteezy.com" websites.
**b** Top: dot plot showing the number of significant pleiotropic associations of HERVs with immune-related diseases across immune cell types. Bottom: Pie charts showing the proportion of diseases detected with significant associations with HERVs in each disease category. **c** Top: bar plots showing the numbers of significant pleiotropic associations of HERVs within each autoimmune disease (left) and cell type (right). Bottom left: heatmap showing the number of cell types found to have associations with each HERV for each autoimmune disease. Bottom middle: heatmap showing the number of diseases associated with each HERV for each cell type. Bottom right: bar plot showing the number of significant pleiotropic associations of each HERV with specific diseases across different cell types. **d** Effect sizes of eQTL SNPs plotted against the allelic effects from the SS GWAS for *MSTD_dup15-chr6* in CD4-T cells. *P*-value is derived from SMR analyses. Multiple testing correction was performed using the Benjamini–Hochberg false discovery rate (FDR) method (threshold = 0.05) implemented via the p.adjust function in R ($n = 20$). The dashed lines represent the estimate of effect size at the top *cis*-eQTL. Error bars are the standard errors of SNP effects. **e** Effect sizes of eQTL SNPs plotted against the allelic

effects from the CD GWAS for *LTR2B_dup15-chr6* in CD4-T cells. *P*-value is derived from SMR analyses. Multiple testing correction was performed using the Benjamini–Hochberg false discovery rate (FDR) method (threshold = 0.05) implemented via the p.adjust function in R ($n = 20$). The dashed lines represent the estimate of effect size at the top *cis*-eQTL. Error bars are the standard errors of SNP effects. **f** LocusZoom plot of CD GWAS (genotype and GWAS association) and *LTR2B_dup15-chr6* eQTLs (genotype and *LTR2B_dup15-chr6* expression association). The y-axis shows the $-\log_{10}(P\text{-value})$ of association tests from GWAS and eQTLs. Points are color-coded based on LD ($r^2$) relative to the variant with the highest colocalization posterior probability in the locus, rs6939196, identified in the colocalization analysis of CD GWAS and *LTR2B_dup15-chr6* eQTLs. **g** Boxplots showing the expression of *LTR2B_dup15-chr6* in normal (left, $n = 6$ biologically independent replicates) and patient (right, $n = 4$ biologically independent replicates) CD4-T cells from CD single-cell data[38]. Individual points represent expression values per donor. Center lines represent medians, box limits indicate the 25th and 75th percentiles, whiskers extend to 1.5× interquartile range (IQR) from the box edges. Statistical significance was assessed by a two-tailed independent samples *t*-test.

to form RNA duplexes, thereby determining enhancer-promoter loops and pairing specificity, which in turn regulate gene expression[43]. Our study demonstrates that HERVs are predominantly enriched in cell-type-specific active genomic regions, such as promoters and enhancers, leading to more specific expression patterns of HERVs across different cell types compared to typical genes. Consequently, HERVs located in promoters and enhancers may also contribute to promoter-enhancer interactions in a similar manner. These findings suggest that HERV expression is primarily regulated by epigenetic mechanisms. Numerous studies have shown that epigenetic profiles exhibit greater divergence between different cell types than transcriptomes[44]. Consistently, we found that HERVs expressed in PBMCs are also more cell type-specific.

HERVs are implicated in various human biological processes, including aging, tumor development, and autoimmune diseases. The immune system serves as a fundamental defense against diseases, particularly those associated with viruses. However, the impact of these viral integrations on our genome and their effects on the immune system remain unclear. In this study, we focused on the expression and regulation of HERVs in PBMCs, finding that HERV expression is cell type-specific. The enrichment of HERVs in cell-type-specific active genomic regions suggests that HERVs may serve as crucial regulatory elements that help maintain cellular states and functions. These findings imply that humans have adapted to these viral elements over the course of evolution, rendering them no longer wild and uncontrolled, but selectively expressed according to the needs of different cell types. By utilizing corresponding genotype data, we found that HERVs possess eQTLs, similar to normal genes. Further integration with disease-associated GWAS data revealed that HERVs may act as mediators between genetic variants and diseases. Thus, our findings indicate that HERVs may play a crucial role in immune-related diseases.

While the SMR analysis provides valuable insights into the potential regulatory relationships between HERVs and autoimmune disease-associated genes, it is important to note that SMR analysis cannot definitively establish causal associations between two phenotypes. Instead, SMR identifies pleiotropic associations, which may arise from shared causal variants. In our study, the observed associations between HERVs and autoimmune disease traits could reflect either direct regulatory effects of HERVs or indirect effects mediated by nearby genes or other genomic elements.

Despite this limitation, the SMR results offer important clues that can guide further experimental validation. For example, the identification of HERVs significantly associated with autoimmune disease traits highlights potential candidate loci for functional studies. Future experiments, such as CRISPR/Cas9-mediated gene editing, chromatin conformation capture (3C), or reporter assays, could be employed to directly test whether HERVs regulate the expression of nearby genes or

influence immune-related pathways. Additionally, integrating multi-omics data (e.g., chromatin accessibility, histone modifications, and transcription factor binding) could provide further mechanistic insights into the regulatory roles of HERVs in autoimmune diseases.

Overall, HERVs exhibit multifaceted roles: on one hand, they serve as indispensable elements and genes in human evolution; on the other hand, when their expression is deregulated, they may contribute to abnormal processes such as aging, tumor development, and autoimmune diseases. Our study provides new insights into the expression and regulatory mechanisms of HERVs in immune cells, enhancing our understanding of the symbiotic evolutionary history between humans and retroviruses, as well as the roles of HERVs in immune-related diseases.

## Methods
### Cohorts
This study complied with all relevant regulations for human participant research and adhered to the principles of the Declaration of Helsinki. The genomic analyses in this study were conducted using exclusively publicly available datasets from GEO (GSE196830[16], GSE157477[38]) and Synapse (syn50209110[26]). The original study for GSE196830 received ethical approval from the Tasmania Health and Medical Human Research Ethics Committee (H0012902), while GSE157477 and syn50209110 were approved by the Institutional Review Boards of Washington University in St. Louis. All original studies obtained appropriate participant informed consent prior to data generation.

### Construction of HERV-gene integrated reference and expression quantification
The HERV annotation file, compiled by RepeatMasker, was obtained from the UCSC Table Browser for GRCh38 (https://genome.ucsc.edu/cgi-bin/hgTables). HERVs that partially overlapped with gene exons were removed to avoid quantitative bias. As a result, 692,759 HERV sites were obtained to provide annotation information for the subsequent quantitative process (Supplementary Data 1). Using CellRanger (v7.1.0), we then constructed a combined reference genome incorporating both filtered HERV annotations and protein-coding genes through the "mkref" function.

PBMC Single-cell RNA-seq data (fastq file) were downloaded from GSE196830[16] and syn50209110[26]. Data processing used CellRanger (v7.1.0) "count" function with these parameters: --fastqs "./fastqs" --sample = "sample"  --localcores  16  --id = "new_matrix"  --transcriptome = "/genome/combined_HERV_gene_reference/"--include-introns = "False". This integrated approach enabled simultaneous profiling of host gene expression and retroelement activity while minimizing cross-mapping artifacts.

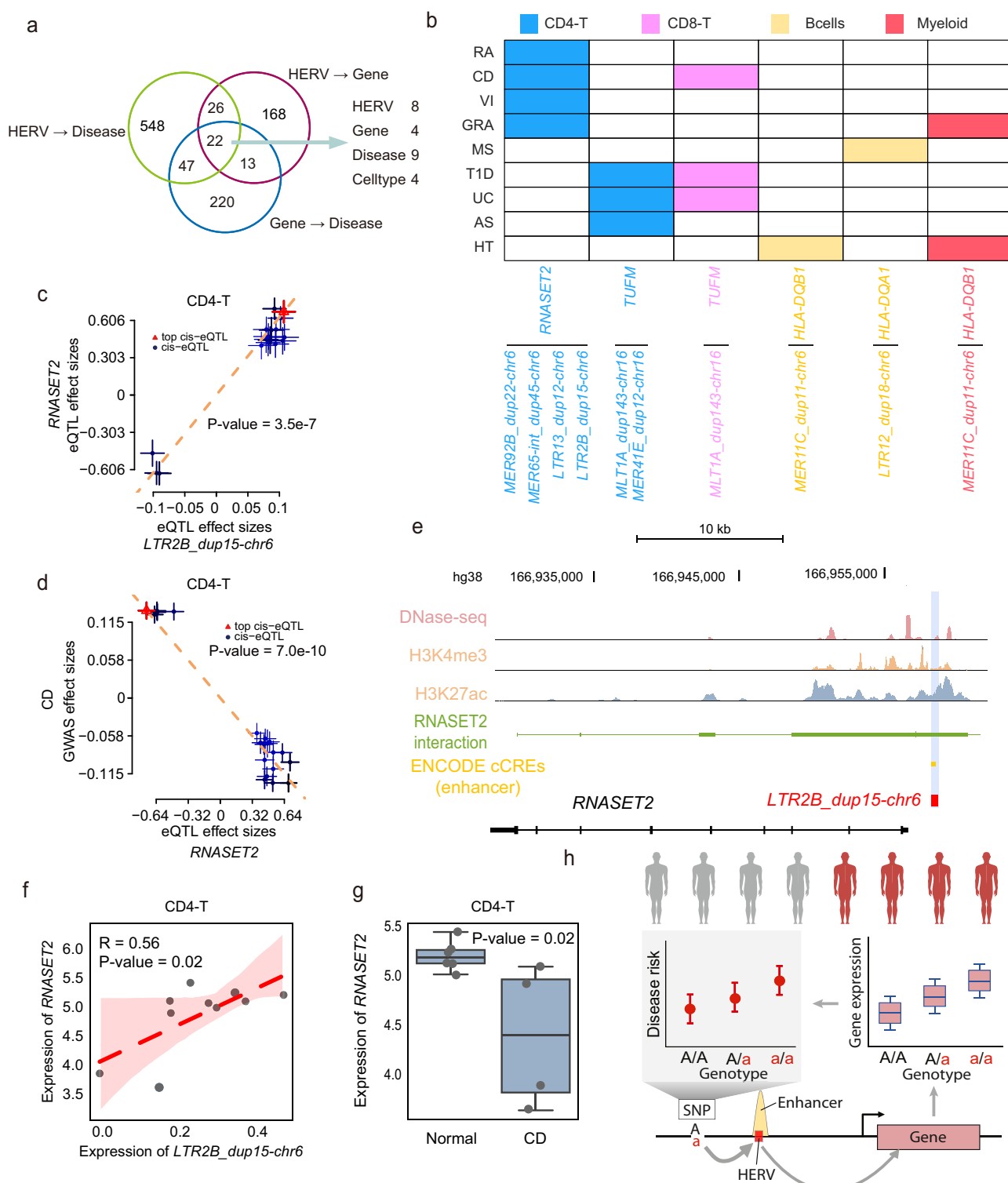

## Single-cell RNA-seq data processing

We then used the Scanpy[45] (version 1.9.8) to perform several QC metrics for each cell, including total counts, total number of genes, and percentage of mitochondrial gene counts. Next, we removed the identified doublets and retained cells within the following QC criteria: $500 <$ total number of genes $< 6000$, $1000 <$ total counts $< 25,000$, and percentage of mitochondrial gene counts $< 10\%$. Additionally, genes expressed in fewer than three cells were excluded.

To account for technical deviations between different samples, such as sequencing depth and batch effects, we used the Counts Per Million (CPM) method for standardization. Specifically, the count of

each HERV was divided by the total counts of the corresponding cell and then multiplied by 1e4. The calculation formula is: (HERV counts*1e4)/(total counts). We only kept HERVs expressed in more than 20 cells for subsequent analysis. We then use the "log1p()" function in scanpy[45] (version 1.9.8) to reduce the skewness of the data distribution.

## Pseudobulk HERV expression

We generated a HERV pseudobulk UMI count matrix for each cell type by extracting the UMI counts of the cell type and aggregating the counts per HERV per individual. The pseudobulk counts were divided

**Fig. 6 | Pleiotropic associations and regulation between HERVs and the expression of disease-associated genes. a** Venn plot showing overlapped pleiotropic associations among HERV-Gene, HERV-Disease and Gene-Disease. **b** Heatmap showing overlapped associations in (**a**). **c** Effect sizes of variants from *LTR2B_dup15-chr6* eQTL plotted against those for variants from the *RNASET2* eQTL in CD4-T cells. *P*-value is derived from SMR analyses. Multiple testing correction was performed using the Benjamini–Hochberg false discovery rate (FDR) method (threshold = 0.05) implemented via the p.adjust function in R ($n = 19$). The dashed lines represent the estimate of effect size at the top *cis*-eQTL. Error bars are the standard errors of SNP effects. **d** Effect sizes of variants from *RNASET2* eQTL plotted against those from the CD GWAS in CD4-T cells. *P*-value is derived from SMR analyses. Multiple testing correction was performed using the Benjamini–Hochberg false discovery rate (FDR) method (threshold = 0.05) implemented via the p.adjust function in R ($n = 20$). The dashed lines represent the estimate of effect size at the top *cis*-eQTL. Error bars are the standard errors of SNP effects. **e** Chromatin conformation data of *RNASET2* and *LTR2B_dup15-chr6* with significant pleiotropic association. H3K27ac, H3K4me3 ChIP-seq, and DAase-seq data for CD4-T cells were downloaded from ENCODE[25]. The *x*-axis denotes the physical position along a segment of chromosome 19 containing the *RNASET2* gene and *LTR2B_dup15-chr6*. *RNASET2*-interaction genome regions are derived from GeneHancer[35], which consists of clustered interactions of GeneHancer regulatory elements and genes. ENCODE cCREs represent the candidate *cis*-Regulatory Elements derived from ENCODE[25]. The exon structure of *RNASET2* is presented in the bottom horizontal track. **f** Scatter plot illustrating the Pearson's correlation between *RNASET2* and *LTR2B_dup15-chr6* expression in CD4-T cells. The dashed line represents the least-squares linear regression fit centered on the conditional mean response. The shaded area indicates the 95% confidence interval (CI) for the mean predicted response at each *x*-value, calculated from the regression standard error. Pearson's correlation coefficient $R = 0.52$ (two-tailed *p*-value = 0.02, calculated by two-sided linear regression and Pearson correlation tests). **g** Boxplots showing the expression of *RNASET2* in normal (left, $n = 6$ biologically independent replicates) and patient (right, $n = 4$ biologically independent replicates) CD4-T cells from CD single-cell data. Individual points represent expression values per donor. Center lines represent medians, box limits indicate the 25th and 75th percentiles, whiskers extend to 1.5× interquartile range (IQR) from the box edges. Statistical significance was assessed by a two-tailed independent samples *t*-test. **h** Schematic illustration of the potential regulatory role of HERVs in diseases. The materials in the pictures were downloaded from "Vecteezy.com" websites.

by the total UMI count of the corresponding cell type and multiplied by 1e6. We then use the "log1p()" function in scanpy[45] (version 1.9.8) to reduce the skewness of the data distribution.

## Batch correction and cell clustering of gene and HERV expression

We utilized the Scanpy[45] (version 1.9.8) workflow for batch effect correction, dimensionality reduction, and cell clustering of both gene and HERV expression matrices. First, we use the default parameters of the "sc.pp.highly_variable_genes()" function to extract highly variable genes/HERV. Then, we applied the "sc.pp.scale()" and "sc.tl.pca()" functions to scale and reduce the dimension of the highly variable gene/HERV matrix. Batch correction was performed using the Harmony algorithm with the "sc.external.pp.harmony_integrate()" function, specifying batch = 'pool'. Subsequently, we computed neighbors on the first 30 Harmony dimensions using the "sc.pp.neighbors()" function and conducted UMAP dimensionality reduction with "sc.tl.umap()". Clustering analysis was performed using "sc.tl.leiden()" with a resolution of 0.3.

For the gene expression data, we performed detailed annotation based on key marker genes and differentially expressed genes between clusters. Major cell lineages were assigned to each cluster of cells using the abundance of canonical marker genes: CD4-T cells (CD3D and CD4), CD8-T cells (CD3D and CD8A), γδ T cells (TRDC), B cells (MS4A1), NK cells (NCAM1), Myeloid cells (CD68), plasma cells (JCHAIN), megakaryocytes (PPBP), and hematopoietic stem cells (CD34).

## HERV and nearest gene expression correlation analysis

Spearman's rank correlation tests were performed using the "scipy.stats" module in SciPy (version 1.12.0) to evaluate the correlation between each expressed HERV and its nearest genes.

## Track production and display

Cell-type-specific bam files of single-cell RNA-seq data were extracted from the output total bam file of cellranger according to the cell barcode of each read. Then BigWig files were generated from cell type-specific bam files using the "bamCoverage" program from deepTools[46] (version 1.12.0) with the following parameters: "--binSize 1", "--normalizeUsing RPKM", "--exactScaling", "--minMappingQuality 10". Cell type-specific BigWig files of H3K27ac, H3K4me3 ChIP-seq, and DNase-seq data were obtained from ENCODE[25]. These BigWig files were then uploaded to the UCSC Genome Browser[47] (https://genome.ucsc.edu/cgi-bin/hgCustom) for visualization.

## Identification of highly variable HERVs and UMAP visualization

Highly variable HERVs ($n = 2045$) were identified using the "sc.pp.highly_variable_genes()" function in Scanpy (v1.9.8) with parameters: min_disp = 0.5, min_mean = 0.1, and max_mean = 4.

Prior to UMAP visualization, we subset the dataset by removing rare cell populations (HSCs, plasma cells, Megakaryocytes, and γδ T cells), retaining only the five major immune cell types (CD4-T, CD8-T, NK, B, and Myeloid cells) with sufficient cell counts. UMAP dimensionality reduction was using "sc.tl.umap()" function in Scanpy (v1.9.8) with parameters: n_neighbors = 40, n_pcs = 5, res = 0.5. Additionally, the cell labels in the figure are derived from annotations based on highly variable genes.

## Differential expression analysis

Cell type-specific expressed HERVs were identified using the "sc.rank_genes_groups()" function in Scanpy (version 1.9.8). HERVs were considered statistically significant if their Bonferroni-adjusted *P*-values were less than 1e−5 and the log2 fold change (log2FC) was >2.

## Cell type-specific HERV enrichment analysis

Fifteen categories of chromatin states for five cell types were downloaded from the FTP site of the Roadmap Epigenome project (https://egg2.wustl.edu/roadmap/web_portal/chr_state_learning.html#core_15state)[24]. Enhancers for each cell type were extracted from the "Genic enhancers" and "Enhancers" categories within the 15 chromatin states corresponding to that cell type. Promoters for each cell type were extracted from the "Active TSS" and "Flanking Active TSS" categories within the 15 chromatin states corresponding to that cell type. The HERVs tested in the differential analysis were divided into four groups based on whether they were cell type-specific HERVs and whether they were in the chromatin state of interest. A contingency table was constructed using the number of HERVs in the four categories. The log odds ratios (ORs) and *P*-values of the two-sided Fisher's exact test were determined using the R package epitools (version 0.5-10.1).

## Generation of genotypes

The methods for generating genotypes were based on the original OneK1K[16] study with slight modifications. Specifically, the genomic coordinate version in the Illumina Infinium Global Screening Array Manifest File was updated from GRCh37 to GRCh38. The Illumina Infinium Global Screening Array Manifest file for the GRCh38 version can be downloaded from: https://support.illumina.com/array/array_kits/infinium-global-screening-array/downloads.html. The remaining steps were performed as in the original study.

## Lineal model-based eQTL detection

We employed TensorQTL[28] (version 1.0.8) to conduct linear model-based cis-eQTL analysis. Briefly, HERVs present in more than 10% of samples within each cell type were included. The remaining pseudo-bulk HERVs expression data were used as phenotype inputs in TensorQTL. The covariates included sex, age, six genotype-based principal components, and two PEER factors. PEER factors were derived from the top 2000 highly variable HERVs. For cis-eQTL analysis, we focused on variants located within 1 Mb upstream or downstream of the HERV's transcription start site (TSS). We used the "map_nominal()" function to obtain the nominal P-values for each variant-HERV pair. Subsequently, we used the "map_cis()" function to perform 10,000 permutations and generate phenotype-level summary statistics with empirical P-values, enabling calculation of genome-wide FDR (q-value). Finally, we used the "cis.map_independent()" function to identify the conditionally independent eQTLs.

## eQTL Functional enrichment analysis

To compare the eQTL variants to a null distribution of similar variants without regulatory association, we sampled for each eQTL variant 100 random regulatory genetic variants matching for relative distance to TSS (within 2.5 kb) and minor allele frequency (within 2%) and only kept variants that are not eQTLs for any other HERV (nominal P-value > 0.05). The enrichment for a given category was calculated as the proportion of the number of regulatory associations in a given category and all regulatory variants over the same proportion in the null distribution of variants. The P-value for this enrichment is calculated with Fisher's exact test. Finally, we corrected for multiple testing using an FDR threshold of 5% using the "p.adjust" function in the R programming language. The code for performing the functional enrichment analysis can be accessed here: https://github.com/NLykoskoufis/fenrichcpp.

## SMR analysis

The eQTL summary statistics for eHERV with nominal P-values were extracted and formatted as BESD files. Additionally, GWAS summary statistics for the 81 immune-related diseases (Supplementary Data 5) and public gene eQTL results[39] from the European population were utilized in the SMR (version 1.3.1) analysis. We employed SMR to detect three types of pleiotropic associations within each cell type: (1) variant-eHERV-eGene, (2) variant-eHERV-disease, and (3) variant-eGene-disease. The default threshold of 5e−8 was used to select the associated eQTLs as instrumental variables for the SMR test. In each cell type, we applied the Bonferroni method to correct P_smr in the SMR results and reject associations with a pHEIDI < 0.01. Associations with a q-value < 0.05 and a pHEIDI > 0.01 were considered significant pleiotropic associations.

## Colocalization analysis of GWAS and eQTLs

To orthogonally verify the Mendelian randomization results between GWAS loci and cis-eQTLs, we tested colocalization between GWAS and eQTL signals for genes with at least ten variants using the "coloc.abf()" function of the coloc R package v.5.2.3[48] (with default prior, using beta coefficients from the GWAS and eQTL analysis). The colocalization results were visualized using LocusZom[49] (http://locuszoom.org/).

## Reporting summary

Further information on research design is available in the Nature Portfolio Reporting Summary linked to this article.

## Data availability

Single-cell gene expression and genotype data from the OneK1K cohort are available through the Gene Expression Omnibus (https://www.ncbi.nlm.nih.gov/geo/query/acc.cgi?acc=GSE196830). The validation PBMC single-cell RNA-seq dataset can be obtained from Synapse (https://www.synapse.org/Synapse:syn50209110). The 15 chromatin state predictions for each immune cell type can be downloaded from the Roadmap Epigenomics Project (https://egg2.wustl.edu/roadmap/web_portal/chr_state_learning.html#core_15state). Histone ChIP-seq (H3K27ac, H3K4me3) and DNase-seq for five immune cell types are available through the ENCODE project (https://www.encodeproject.org/). Summary statistics for 81 diseases (Supplementary Data 5) can be downloaded from the GWAS Catalog (https://www.ebi.ac.uk/gwas/). Single-cell gene expression data for CD are available through the Gene Expression Omnibus (https://www.ncbi.nlm.nih.gov/geo/query/acc.cgi?acc=GSE157477). Source data are provided with this paper.

## Code availability

All software used in the study is publicly available as described in the "Methods" section. The custom code for all analyses presented in this study is available on GitHub (https://github.com/YoungLi88/HERV_eQTL) and Zenodo (https://doi.org/10.5281/zenodo.15966381)[50].

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

## Acknowledgements

We gratefully acknowledge the support from the Zhejiang Science and Technology Department (No. 2024C03004). This work was also supported by the Zhejiang Key Laboratory of Spatial Omics. We would like to thank DCS Cloud (https://cloud.stomics.tech) for providing the computational resources and software support necessary for this study.

## Author contributions

F.Z., Young Li, and Yi Liu conceived of and designed the study. F.Z., Yi Liu, J.H.L., X.X.L., Z.X.J., and Young Li designed and performed the computation and statistical analyses. X.D. and Young Li acquired the funding. X.D. Y.G. and Young Li supervised the study. F.Z., Yi Liu, X.X.L., J.H.L., Z.X.J., and Young Li wrote, reviewed and edited the manuscript.

## Competing interests

The authors declare no competing interests.
