## [Transparent Peer Review file · Nature Communications]

Single-Cell eQTL Mapping of Human Endogenous Retroviruses Reveals Cell Type-Specific Genetic Regulation in Autoimmune Diseases

Corresponding Author: Dr Young Li

Version 0:

Reviewer comments:

Reviewer #1

(Remarks to the Author)

The role of human endogenous retroviruses (HERVs) in gene regulation and their impact on human diseases remains poorly understood. To further characterize this relationship, Zhu et al. analyzed publicly available data to map the contribution of HERVs on human disease. First, they quantified cell-type-specific HERV expression patterns in a publicly available PBMC single-cell RNA-seq dataset, while also identifying genetic mutations associated with differential HERV expression (HERV cis-eQTLs). Leveraging these discovered QTLs, they used eQTL and GWAS summary statistics to decompose causal contributions to autoimmune traits. Overall, the study was thoughtfully executed and well done, highlighting a potential contributor to complex traits that have been understudied. However, I have some important concerns regarding the processing pipeline and the analyses of SMR results that should be addressed prior to publication.

Major

- My understanding is that HERV elements are highly repetitive, which can be challenging for short read-mapping algorithms. Unfortunately, minor artifacts during the raw data processing stages can sometimes influence downstream results. To ensure the reliability of the findings, I need further reassurance that reads were mapped appropriately to avoid issues such as repeat counting of repetitive HERVs or other common artifacts associated with repetitive elements. Specifically, how exactly was the read mapping performed? Was a single index constructed that included both the transcriptome and HERV elements, allowing reads to be jointly mapped? Or was a two-step process used?

In such scenarios, I typically rely on reviewing the code to understand the exact steps used to perform the analysis. However, I did not find any code available related to this aspect of the analysis. It would be helpful if the authors shared some version of the processing pipeline they used. Additionally, how does this approach compare to other approaches that have been used in similar studies?

- I appreciate the SMR analysis; however, the authors should address alternative explanations for the observed associations before conclusively attributing a role for HERVs in autoimmune diseases. For example, the authors propose that SNPs impact HERV expression, which in turn influences autoimmune traits. However, could it be that SNPs associated with HERV expression also impact nearby genes which are the actual mediators of the autoimmune disease effect?

Similarly, I would like the authors to address the analysis linking eHERVs to nearby gene expression. Is it possible that a SNP impacts both HERV and nearby gene expression independently, rather than the HERV mediating the change in gene expression? At the very least, the authors should discuss evidence for these alternative causal pathways and consider their implications for the interpretation of the results.

Minor

- Please include all the numbers of the sets in the Venn diagram plot in Fig 5a.

(Remarks on code availability)

I did not run the code, but it looks helpful. My one concern is that there isn't code for how the read mapping and other processing steps were performed. I've included my concern in my comments to the authors.

Reviewer #2

(Remarks to the Author)

This work investigates the expression and disease association of HERVs in human PBMCs using publicly available single cell RNASeq data. The authors map HERVs in the OneK1K dataset using a custom GTF to define their expression patterns across common immune cell populations and identify a large number of expressed loci. They characterize both cell type specific expression patterns amongst the identified HERVs and go on to identify cis-eQTLs associated with these loci. Through the identification of HERV associated eSNPs, the authors investigate disease associations with the HERV expression. They find numerous disease-associated eHERVs, with particular enrichment for immune disease, especially autoimmune digestive diseases. Finally, they describe a HERV associated with the expression of a gene linked strongly to celiac disease severity, suggesting that the HERV may contribute to celiac disease susceptibility through regulation of this intermediate gene.

Comments

HERVs reported in this manuscript include any HERV detected in a minimum of 20 cells. For a dataset of greater than one million cells total, this is a very lenient filter to apply. It would be constructive if the authors could justify this cutoff in either the body of the text or the methods. Additionally, a plot should be included demonstrating the number of cells a given HERV is detected in. If the majority of HERVs are consistently detected at very low frequencies in very few cells, this may explain some degree of the described cell type specific effects reported throughout the manuscript.

The language in this manuscript should be softened in regards to causality. For example, in line 229 the authors state that "increased expression of MER65-int_dup45-chr6 leading to reduced susceptibility to CD ($\beta = -0.81$)". While there is an apparent association between MER65-int_dup45-chr6 expression and CD, there is not sufficient causal evidence that the HERV leads to the disease. A similar claim is made in line 256 and 77. The text should be evaluated for these statements and corrected as appropriate.

The SMR analysis demonstrates a relationship between MER92B_dup23-chr6 as a regulator of RNASET2, both of which are associated with CD. However, the authors do not address if these genes are in linkage disequilibrium or are both eQTL genes of a shared SNP or two linked SNPs. (This is a common assumption as explained in Zhu et al. 2016) It is possible that MER92B_dup23-chr6 and RNASET2 are co-expressed which explains their shared association with CD. There is a need to further justify the model that the HERV regulates RNASET2, as is depicted in figure 5h, or the claim needs to be weakened.

Minor points:

How do the identified cis-eQTLs in this manuscript compare to (She et al. 2022)? The authors should comment on how this work compares to HERV eQTL analysis from bulk sequencing data.

In line 102, can the authors clarify the use of significant reads in the following statement "Additionally, 24547 (43%) expressed HERVs were found in intergenic regions (Fig. 1e), supported by significant sequencing reads, indicating programmed expression rather than sequencing noise (Fig. 1f)."

In Figure 2c, the data is re-clustered using highly variable HERVs, are the cells of other cell types that are no longer reflected in the data removed from the analysis or encompassed by the remaining clusters? If the latter, this should be noted that the clusters are not necessarily reflective of the cell types.

In figures 2c and 2e, the UMAP should be displayed with the individual points visible so that it is possible to infer the sample size.

In figure 2i, the tracks should be extended to include nearby genes that might also be co-regulated.

Supplementary figure 2f is called out in the text but does not exist.

In the abstract, the statement is made that 56668 expressed HERV loci with cell type-specific expression were identified. I do not believe the claim is made in the main text that these loci are all cell-type specific.

Works cited

She, J., Du, M., Xu, Z. et al. The landscape of hervRNAs transcribed from human endogenous retroviruses across human body sites. *Genome Biol* 23, 231 (2022). <https://doi.org/10.1186/s13059-022-02804-w>

Zhu, Z., Zhang, F., Hu, H. et al. Integration of summary data from GWAS and eQTL studies predicts complex trait gene targets. *Nat Genet* 48, 481–487 (2016). <https://doi.org/10.1038/ng.3538>

(Remarks on code availability)

Reviewer #3

(Remarks to the Author)

Single-Cell eQTL Mapping of Human Endogenous Retroviruses Reveals Cell Type-Specific Genetic Regulation in Autoimmune Diseases

Zhu et al provide an impressive bioinformatic characterization of HERV element expression and activity in the PBMCs of individuals and leverage this data to associate with autoimmune diseases. The data are clean, convincing, and well organized. However, there are instances in the text and referencing in which the authors fail to address certain nuances of HERV elements, which can be admittedly difficult. Otherwise, the authors have approached their results without overreaching and sensationalizing, which is of high importance regarding the work conducted. Overall, the manuscript is impactful, timely, and of high-quality. The writing of the text would benefit from some English editing.

Comments

Line 18: It should be made clear that HERVs are not TEs, and that this is a common misconception and term that is frequently, and inappropriately, utilized by the field in the literature. More accurately, they should instead be referred to and classified under “endogenous retroelements” due to their lack of transposition. Please see the following review in example and for clarification DOI: 10.1038/s41580-023-00674-z

Line 39: The structure described would be representative of an intact provirus, which is not the case for the majority of HERV elements. Some more descriptive background on the typical structures of modern ERV elements in the human genome would provide a more descript and accurate introduction.

Line 44: It would be ingenuine to relay that HERVs “evolved” to provide utility, it is more so that the relationship with HERVs and the host have evolved to coopt or utilize their sequences. Please rephrase.

Line 47: This should read “Solo LTRs are the most common form”

Line 47, 50, 51, and throughout: Please standardize the usage of acronyms at their first occurrence (e.g., LTR and IFN, which is later defined at line 51).

Line 54: I may be unaware of some literature, but I am unaware of any evidence showing that the progenitor viruses responsible for endogenization events causes any disease other than retroviremia. While intuitive that their would be comorbidities and such, I do not believe there is any evidence of which. Please rephrase or add citations.

Line 64: other than these 2 examples, there are pretty substantial advancements that have been main regarding HERV activity and immunity that also warrant discussion. Please expand upon this section of the introduction. Furthermore, HERV RTase activity is quite controversial within the field and some of the better described mechanisms (e.g., transcriptional regulation and immunogenic RNA production) warrant discussion as well. DOI: 10.1146/annurev-immunol-101721-033341

Line 91: I agree with this methodology, please just indicate why HERVs overlapping with gene exons were excluded for any readers who are unfamiliar with why this was done.

Figure 1D: I am always quite interested in these sorts of plots and would appreciate a more in-depth demonstration of this by individual and by cell type. It would be incredibly informative to add an additional supplemental figure, which can also be referred to on line 97, that breaks down these gross expression profiles as such, and include statistical testing when appropriate. This bulked analysis is fine for the main text, but how does this differ amongst cell types across individuals?

Figure 2C: I'm impressed by how effective and clean this UMAP projection came out. Can you add a breakdown of how many of these HERVs are intronic vs intergenic?

Line 218: Can you clarify what you mean by this statement regarding an interaction with 8 HLA genes? Similarly, BRD2.

Line 223: This would more be attributed to putative roles of HERVs to impact B cell functionality to contribute to autoinflammation more so than the roles of B cells in autoimmunity. Please rephrase.

Line 242: While the section itself is fine in the language, the heading needs to be rephrased. While this is impressive bioinformatic work, without wet lab validation it would be ingenious. Please soften to “HERVs potentially regulate autoimmune disease-associated genes” or something of that manner.

Line 291: This sentence does not make much sense and is attempting to combine two separate thoughts. Please divide this into two statements that are better relayed.

Line 293: Numerous studies and bioinformatic pipelines have permitted the determination of locus-specific HERV expression from numerous datasets. While issues from multimapping, dropouts, read length, and number of reads from certain NGS datasets make this difficult, the problem has been approached and accomplished numerous times. This statement is both dated and no longer factual. Delete entirely.

Line 304: The way this statement reads makes it seem as if the authors are the first to apply similar methodologies to perform locus-specific quantification of HERV elements from short read scRNA sequencing data, which is simply not the case. Please revise the last statement of this paragraph to better relay that.

Line 306: Similar to my first comment regarding line 18, please ensure that the most correct nomenclature is utilized.

Line 315: I am confused by what the authors mean in “rather than typical gene expression pathways”, as if epigenetics are not part of typical genetic regulation? Please revise this statement to be clearer by what is meant

Line 328: Please delete the sentence beginning on this line. It is not necessary and would likely be a source of contention across multiple camps of people in retrovirology, paleovirology, genetics, etc.

Line 339 and potentially elsewhere: I think it is important to change the language from “.. ... they can facilitate... ..” to something along the lines of “.. ... their deregulated activity may facilitate”. Importantly, the presence of HERV elements alone is not a nefarious detriment to human health, and conditions need to be met for their deleterious effects. This may also require better defining elsewhere in the document as well.

Discussion: Due to the bioinformatic nature of the article, it is incredibly important that the authors provide a paragraph on the limitations of the study. This should be added as the second to last paragraph.

(Remarks on code availability)

Briefly reviewing and I am not a bioinformatician, but the code appears well organized to my naive eyes

Version 1:

Reviewer comments:

Reviewer #1

(Remarks to the Author)

The authors have addressed most of my concerns, and I remain enthusiastic about the study. However, one major issue remains that should be addressed before publication.

The current pipeline quantifies HERV expression by aligning the same reads separately to two references: one for human genes and one for HERVs. This approach introduces a key flaw: reads that appear uniquely mapped in one reference may actually map equally well to loci in the other, but this ambiguity is hidden by performing alignments independently. As a result, HERV expression may be overestimated due to misclassified multi-mapping reads.

This problem is especially relevant for repetitive or homologous sequences like HERVs, and is analogous to known issues in mitochondrial RNA analysis, where reads from NUMTs can be misassigned unless aligned to a combined genome index.

By contrast, tools like CELLO-seq, scTE, and soloTE use a single, comprehensive reference for alignment, followed by post-processing to assign reads. This ensures that mapping ambiguity is properly handled.

To improve mapping accuracy and avoid inflation of HERV expression, the authors should revise their pipeline to align reads once to a combined reference containing both genes and HERVs, and then apply appropriate read attribution.

(Remarks on code availability)

Did not run the code, but pleased that it looks comprehensive.

Reviewer #2

(Remarks to the Author)

The responses and revisions made by the authors have addressed my questions and concerns. I suggest that in the methods or text the authors further clarify that the data was subsetted for visualization and analysis in figure 3c prior to re-clustering.

(Remarks on code availability)

Reviewer #3

(Remarks to the Author)

The authors did a great job at improving the article which is of timely importance and interest. Some minor comments remain.

Line 52 – Remove "flanking". Just LTRs would be fine and it reads like they must flank genes from both ends to impact expression.

Line 58 – the sentences ending and beginning here could be worded more clearly. Consider rephrasing to "where that may contribute to tumorigenesis^{7,8} autoimmunity^{9,10}, and senescence^{aging}^{11,12}."

Figure 1 – what were the proportions of reads denoted as multimapping using the parameters described? I worry this may be a substantial portion of potentially relevant information. I do believe the methodology is stringent and sound as possible. Other approaches typically apply EM or something to try to compensate for this, albeit also not perfectly or with as many or more flaws inherently. It would be good to record in a supplementary table or something the number/proportion of discarded reads per sample in order to convey what could be deemed as missing information to the reader. Apologies if this is already done and I missed it.

Line 116 – In regards to SoloTE, I'm not sure if random assignment is the best phrase to characterize EM-based approaches. Inferred assignment would be much better and sound less biased.

Line 408 – "HERVs in the human genome" reads a bit redundant. Just phrase it as "HERVs" or "ERVs in the human genome"

(Remarks on code availability)

Quick overview at previous stage of review

Version 2:

Reviewer comments:

Reviewer #1

(Remarks to the Author)

My previous concerns have been addressed.

(Remarks on code availability)

Reviewer #3

(Remarks to the Author)

I believe the authors have addressed my concerns, and those of reviewer#1 (in my opinion), appropriately. The article is of good quality.

(Remarks on code availability)

Reviewer #1 (Remarks to the Author)

The role of human endogenous retroviruses (HERVs) in gene regulation and their impact on human diseases remains poorly understood. To further characterize this relationship, Zhu et al. analyzed publicly available data to map the contribution of HERVs on human disease. First, they quantified cell-type-specific HERV expression patterns in a publicly available PBMC single-cell RNA-seq dataset, while also identifying genetic mutations associated with differential HERV expression (HERV cis-eQTLs). Leveraging these discovered QTLs, they used eQTL and GWAS summary statistics to decompose causal contributions to autoimmune traits. Overall, the study was thoughtfully executed and well done, highlighting a potential contributor to complex traits that have been understudied. However, I have some important concerns regarding the processing pipeline and the analyses of SMR results that should be addressed prior to publication.

Major

- My understanding is that HERV elements are highly repetitive, which can be challenging for short read-mapping algorithms. Unfortunately, minor artifacts during the raw data processing stages can sometimes influence downstream results. To ensure the reliability of the findings, I need further reassurance that reads were mapped appropriately to avoid issues such as repeat counting of repetitive HERVs or other common artifacts associated with repetitive elements. Specifically, how exactly was the read mapping performed? Was a single index constructed that included both the transcriptome and HERV elements, allowing reads to be jointly mapped? Or was a two-step process used?

Reply: We sincerely thank the reviewer for their insightful comment regarding the challenges of mapping reads to highly repetitive HERV elements. Here, we provide a detailed response to address this concern:

(1) Read Mapping Strategy:

In our study, we employed a two-step independent mapping approach to ensure accurate and reliable alignment of reads to both human genes and HERV elements. Specifically: Step 1: We downloaded a GTF file for human genes and aligned the reads to this reference using cellranger (version 7.1.0), which generated a gene expression matrix.

The alignment script used for processing the data is as follows:

```

cellranger count \
--fastqs /fastqs/ \
--sample= "sample_name" \
--localcores 16 \
--nosecondary \
--id=out_sample_gene \
--transcriptome= "/genome/gene/"

```

Step 2: We constructed a separate GTF file for HERV elements and aligned the same set of reads to this reference independently. It should be pointed out that we excluded HERV loci overlapping gene exons to minimize confounding effects from human gene transcription signals and to focus on HERVs with potential independent regulatory roles, which generated a HERV expression. The alignment script used for processing the data is as follows:

```

cellranger count \
--fastqs /fastqs/ \
--sample= "sample_name" \
--localcores 16 \
--nosecondary \
--id=out_sample_herv \
--transcriptome= "/genome/HERV/"

```

This approach allowed us to optimize the alignment parameters for host genes and HERV elements separately, ensuring high accuracy for both.

(2) Handling Repetitive HERV Elements:

To address the high repetitiveness of HERVs, we configured cellranger to retain only unique mapping reads. (**Fig. 1c**) This step minimized the impact of repeat counting and other artifacts associated with repetitive elements.

Fig. 1c Strategies for quantifying HERVs.

We further assessed the proportion of unique reads for each HERV and found that most HERVs exhibited a proportion close to 100% (**Supplementary Fig. 1a**), indicating that HERV expression was predominantly driven by unique reads.

Supplementary Fig. 1a The kernel density plot illustrates the distribution of the percentage of unique reads mapped to HERVs.

For example, in a single cell, in the case of *Harlequin-int_dup64-chr1* and *Harlequin-int-chr17*, a total of 4 and 5 reads were initially mapped to these two HERVs, respectively. Among these, one read was mapped to both HERVs. After filtering out multi-mapping reads, only 3 and 4 uniquely mapped reads remained, demonstrating the effectiveness of our approach in handling repetitive elements. (**Fig. 1d**)

Fig. 1d Example of read filtering in a single cell

By retaining only unique mapping reads and using a robust alignment pipeline, we ensured the reliability of our findings and minimized the impact of potential artifacts associated with repetitive HERV elements.

We have added the above details to the **Methods** and **Results** section of the revised manuscript to provide a comprehensive description of our read mapping strategy and the steps taken to address the challenges of mapping reads to repetitive HERV elements.

In such scenarios, I typically rely on reviewing the code to understand the exact steps used to perform the analysis. However, I did not find any code available related to this

aspect of the analysis. It would be helpful if the authors shared some version of the processing pipeline they used.

Additionally, how does this approach compare to other approaches that have been used in similar studies?

Reply: We fully agree with the reviewer on the importance of transparency and reproducibility. We have updated our github repository (https://github.com/YoungLi88/HERV_eQTL) and added this part of the code in “mapping” directory.

Regarding the comparison with other approaches used in similar studies, we provide the following insights:

Method	References	Mapping strategies	Throughput	Data Requirements	Resolution
Our Approach	DOI: 10.1016/j.ebiom.2022.104319	Retain only uniquely mapped reads	High (10000+ cell)	short-read	Locus
soloTE	DOI: 10.1038/s42003-022-04020-5	Randomly assigns multi-mapping reads to HERV loci	High (10000+ cell)	short-read	Locus
scTE	DOI: 10.1038/s41467-021-21808-x	Aggregates expression across all instances of a specific HERV family (e.g., LTR12C, MLT1A).	High (10000+ cell)	short-read	Family
CELLO-Seq	DOI: 10.1038/s41587-021-01093-1	Integrates long-read sequencing with a bespoke computational framework.	Low 6-96 cells	Long-read	Locus

The study published in EBioMedicine (DOI: 10.1016/j.ebiom.2022.104319) employed a similar approach to quantify HERV expression at the locus level. Like our method, they used unique mapping reads to ensure accurate and reliable quantification of individual HERV elements. This consistency in methodology further supports the robustness and validity of our approach.

We have added the above details to the results section of the revised manuscript.

“Our approach to quantifying HERV expression at the locus level overcomes key limitations of existing methods. Unlike scTE¹ (family-level aggregation) and soloTE² (random assignment of multi-mapping reads), which either lack locus-specific resolution or introduce bias, and CELLO-Seq³, which is limited to small-

scale datasets, our method retains only unique mapping reads, ensuring high accuracy and scalability (Supplementary Fig. 1b). This approach has been validated in similar studies⁴, further supporting its robustness and applicability.”

- I appreciate the SMR analysis; however, the authors should address alternative explanations for the observed associations before conclusively attributing a role for HERVs in autoimmune diseases. For example, the authors propose that SNPs impact HERV expression, which in turn influences autoimmune traits. However, could it be that SNPs associated with HERV expression also impact nearby genes which are the actual mediators of the autoimmune disease effect?

Reply: We sincerely thank the reviewer for their insightful comment regarding the potential alternative explanations for the observed associations between HERVs and autoimmune diseases. We acknowledge the reviewer’s point that SNPs associated with HERV expression could also impact nearby genes, which might be the actual mediators of the autoimmune disease effect. To address this possibility, we examined whether the SNPs associated with eHERV expression also influence the expression of nearby genes (within a 1 Mb window). The results showed that the majority of SNPs (91.4%) associated with HERV expression are not associated with the expression of nearby genes, suggesting that the observed SNP-HERV-disease associations are unlikely to be driven by nearby genes.

Proportion of HERV-Associated SNPs With/Without Gene eQTLs

We greatly appreciate the reviewer’s attention to this important issue. Their feedback has helped us strengthen the clarity and rigor of our manuscript.

Similarly, I would like the authors to address the analysis linking eHERVs to nearby gene expression. Is it possible that a SNP impacts both HERV and nearby gene

expression independently, rather than the HERV mediating the change in gene expression? At the very least, the authors should discuss evidence for these alternative causal pathways and consider their implications for the interpretation of the results.

Reply: We sincerely thank the reviewer for their insightful comment regarding the potential alternative explanations for the observed associations between eHERVs and nearby gene expression.

According to the explanation in Zhu et al. 2016 (DOI: 10.1038/ng.3538), this model (SMR and HEIDI) cannot distinguish between causal and pleiotropic associations.

[editorial note: figure redacted]

Three possible explanations for an observed association between a trait and gene expression through genotypes. This figure is from Zhu et al. 2016.

Therefore, we acknowledge the reviewer's point that a SNP may independently impact both eHERV and nearby gene expression, rather than the eHERV mediating the change in gene expression. This alternative causal pathway is supported by several potential mechanisms:

(1) Shared Regulatory Variants

A SNP located in a regulatory region (e.g., enhancer or promoter) could independently influence both the HERV and a nearby gene. For example, a single SNP might alter chromatin accessibility or transcription factor binding, thereby affecting the expression of both the HERV and the adjacent gene.

(2) Pleiotropic Effects

A SNP might have pleiotropic effects, simultaneously impacting multiple genomic elements (e.g., HERVs and genes) through distinct molecular mechanisms. This could result in co-expression of the HERV and the nearby gene without a direct regulatory relationship.

To address this, we have revised the title of Figure 6 from "HERVs regulate autoimmune disease-associated genes" to "HERVs **potentially** regulate autoimmune disease-associated genes."

Additionally, we have **softened the language** throughout the manuscript to more accurately reflect the associative nature of our findings rather than implying direct causality.

Line 229: The revised text now reads:

“For example, we found that *MER65-int_dup45-chr6* was negatively correlated with CD only in CD4-T cells, with increased expression of *MER65-int_dup45-chr6* **associated with** reduced susceptibility to CD ($\beta = -0.81$) (Fig. 5e).”

Line 255: The revised text now reads:

“Additionally, we noted that a single HERV locus **could be linked to** the regulation of multiple genes regulate multiple genes, ranging from 1 to 4 (Supplementary Fig. 8b).”

Line 256: The revised text now reads:

“For instance, *LTR12C_dup55-chr5* **is associated with** the regulation of *TRIM23* and *MENPK* exclusively in CD4-T cells (Supplementary Fig. 8c), while *MER4D1_dup36-chr5* **is linked to** the regulation of *DIP2A* in both CD4-T and CD8-T cells (Supplementary Fig. 8d).”

Line 77: The revised text now reads:

“We further illustrated that one potential mechanism **underlying** the HERV-disease associations **could be** that HERVs **may influence the expression** of autoimmune disease-related genes.”

As stated in the article, we investigated the epigenetic relationship between *MER92B_dup23-chr6* and *RNASET2*. We integrated H3K27ac, H3K4me3 ChIP-seq, and ATAC-seq data of CD4+ T cells from ENCODE. We found significant enhancer signals at the *MER92B_dup23-chr6* locus, located 2 kb upstream of *RNASET2* (Fig. 6e). Moreover, the GeneHancer database documents interactions between this enhancer and *RNASET2* (Fig. 6e). These data suggest that *MER92B_dup23-chr6* **may** plays a critical role in regulating the expression of *RNASET2*.

We have also added a discussion of these alternative causal pathways in the Discussion section. The revised text now reads:

“**While the SMR analysis provides valuable insights into the potential regulatory relationships between HERVs and autoimmune disease-associated genes, it is important to note that SMR cannot definitively establish causal associations between two phenotypes. Instead, SMR identifies pleiotropic associations, which may arise from shared causal variants. In our study, the observed associations between HERVs and autoimmune disease traits could reflect either direct regulatory effects of HERVs or indirect effects mediated by nearby genes or other genomic elements.**

Despite this limitation, the SMR results offer important clues that can guide further experimental validation. For example, the identification of HERVs significantly associated with autoimmune disease traits highlights potential

candidate loci for functional studies. Future experiments, such as CRISPR/Cas9-mediated gene editing, chromatin conformation capture (3C), or reporter assays, could be employed to directly test whether HERVs regulate the expression of nearby genes or influence immune-related pathways. Additionally, integrating multi-omics data (e.g., chromatin accessibility, histone modifications, and transcription factor binding) could provide further mechanistic insights into the regulatory roles of HERVs in autoimmune diseases.”

We believe these revisions provide a more balanced interpretation of our results and highlight the importance of future studies to confirm the regulatory mechanisms underlying the observed associations.

Minor

- Please include all the numbers of the sets in the Venn diagram plot in Fig 5a.

Reply: Thank you for your valuable suggestion. We have now included all the numerical values in the Venn diagram.

Fig. 6a Venn plot showing Overlapped pleiotropic associations among HERV-Gene, HERV-Disease and Gene-Disease.

(Remarks on code availability)

I did not run the code, but it looks helpful. My one concern is that there isn't code for how the read mapping and other processing steps were performed. I've included my concern in my comments to the authors.

Reply: We sincerely thank the reviewer for their valuable feedback and for highlighting the need for clarity regarding the read mapping and processing steps.

In response to the reviewer's concern, we have updated our GitHub repository (https://github.com/YoungLi88/HERV_eQTL) to include detailed code for the read mapping and other processing steps. These scripts are now available in the “mapping”,

“characterization_of_HERVs” and **“cell type-specific”** directories of the repository. We believe this addition will significantly enhance the reproducibility of our analysis and serve as a comprehensive resource for researchers interested in replicating or extending our work.

Reviewer #2 (Remarks to the Author):

This work investigates the expression and disease association of HERVs in human PBMCs using publicly available single cell RNA-Seq data. The authors map HERVs in the OneK1K dataset using a custom GTF to define their expression patterns across common immune cell populations and identify a large number of expressed loci. They characterize both cell type specific expression patterns amongst the identified HERVs and go on to identify cis-eQTLs associated with these loci. Through the identification of HERV associated eSNPs, the authors investigate disease associations with the HERV expression. They find numerous disease-associated eHERVs, with particular enrichment for immune disease, especially autoimmune digestive diseases. Finally, they describe a HERV associated with the expression of a gene linked strongly to celiac disease severity, suggesting that the HERV may contribute to celiac disease susceptibility through regulation of this intermediate gene.

Comments

HERVs reported in this manuscript include any HERV detected in a minimum of 20 cells. For a dataset of greater than one million cells total, this is a very lenient filter to apply. It would be constructive if the authors could justify this cutoff in either the body of the text or the methods. Additionally, a plot should be included demonstrating the number of cells a given HERV is detected in. If the majority of HERVs are consistently detected at very low frequencies in very few cells, this may explain some degree of the described cell type specific effects reported throughout the manuscript.

Reply: We thank the reviewer for their valuable comment regarding the filtering threshold for HERVs in our study. In response to the reviewer's suggestion, we have added a detailed explanation and supplementary figure (Supplementary Fig. 1c) to the results section of the revised manuscript. Due to the overall low expression frequency of HERVs, we chose a lenient threshold of >20 cells to retain as many HERVs as possible for downstream analysis. This approach ensures that we do not miss potentially biologically relevant HERVs that may play a role in cell type-specific regulation or disease associations, even if they are expressed at low levels. This threshold strikes a balance between capturing meaningful signals and minimizing the inclusion of technical noise or artifacts. As shown in the figure below (now included in the results section), while the filtered HERVs are detected in a relatively small number of cells

(typically 20–100 cells), the cell type-specific HERVs reported in our manuscript are expressed in a significantly higher number of cells, often ranging from 1,000 to 10,000 cells. This indicates that the cell type-specific effects we describe are driven by HERVs expressed at higher frequencies within specific cell populations, rather than low-frequency artifacts.

Supplementary Fig 1c The kernel density plot illustrates the distribution of HERVs at three stages: Raw HERVs (initial unfiltered data), Filtered HERVs (filtering out HERVs expressed in fewer than 20 cells), and Cell type-specific HERVs

The revised text now reads:

“Due to the overall low expression frequency of HERVs, we applied a lenient threshold of >20 cells to retain as many HERVs as possible for downstream analysis (Supplementary Fig. 1c). This approach ensures that we do not miss potentially biologically relevant HERVs that may play a role in cell type-specific regulation or disease associations. After filtering out HERVs expressed in fewer than 20 cells, we identified 56668 expressed HERV loci in PBMCs (Supplementary Table 2)”

“..... As shown in Supplementary Fig. 1c, while the filtered HERVs are detected in a relatively small number of cells (typically 20–100 cells), the cell type-specific HERVs reported in our manuscript are expressed in a significantly higher number of cells, often ranging from 1,000 to 10,000 cells. This indicates that the cell type-specific effects we describe are driven by HERVs expressed at higher frequencies within specific cell populations, rather than low-frequency artifacts”

The language in this manuscript should be softened in regards to causality. For example, in line 229 the authors state that “increased expression of MER65-int_dup45-chr6 leading to reduced susceptibility to CD ($\beta = -0.81$)”. While there is an apparent association between MER65-int_dup45-chr6 expression and CD, there is not sufficient

causal evidence that the HERV leads to the disease. A similar claim is made in line 256 and 77. The text should be evaluated for these statements and corrected as appropriate.

Reply: We sincerely thank the reviewer for their valuable feedback regarding the language used in our manuscript, particularly in relation to claims of causality. We agree that the language should be softened to more accurately reflect the associative nature of our findings rather than implying direct causality.

Line 229: The revised text now reads:

“For example, we found that *MER65-int_dup45-chr6* was negatively correlated with CD only in CD4-T cells, with increased expression of *MER65-int_dup45-chr6* **associated with** reduced susceptibility to CD ($\beta = -0.81$) (Fig. 5e).”

Line 242: The revised heading now reads:

“HERVs **Potentially** Regulate Autoimmune Disease-Associated Genes”

Line 255: The revised text now reads:

“Additionally, we noted that a single HERV locus **could be linked to** the regulation of multiple genes regulate multiple genes, ranging from 1 to 4 (Supplementary Fig. 8b).”

Line 256: The revised text now reads:

“For instance, *LTR12C_dup55-chr5* **is associated with** the regulation of *TRIM23* and *MENPK* exclusively in CD4-T cells (Supplementary Fig. 8c), while *MER4D1_dup36-chr5* **is linked to** the regulation of *DIP2A* in both CD4-T and CD8-T cells (Supplementary Fig. 8d).”

Line 77: The revised text now reads:

“We further illustrated that one potential mechanism **underlying** the HERV-disease associations **could be** that HERVs **may influence the expression** of autoimmune disease-related genes.”

The SMR analysis demonstrates a relationship between *MER92B_dup23-chr6* as a regulator of *RNASET2*, both of which are associated with CD. However, the authors do not address if these genes are in linkage disequilibrium or are both eQTL genes of a shared SNP or two linked SNPs. (This is a common assumption as explained in Zhu et al. 2016) It is possible that *MER92B_dup23-chr6* and *RNASET2* are co-expressed which explains their shared associated with CD. There is a need to further justify the model that the HERV regulates *RNASET2*, as is depicted in figure 5h, or the claim needs to be weakened.

Reply: We sincerely thank the reviewer for your insightful comment regarding the

relationship between *MER92B_dup23-chr6* and *RNASET2* in the context of CD. Here, we provide a detailed response to address this concern:

In our study, we performed Summary-data-based Mendelian Randomization (SMR) analysis to investigate the potential regulatory relationship between *MER92B_dup23-chr6* and *RNASET2*. To rule out the possibility of linkage disequilibrium (LD) or pleiotropy, we conducted the HEIDI (Heterogeneity in Dependent Instruments) test. The HEIDI test assumes only one causal variant in the cis-eQTL region, and its results ($p > 0.01$) indicated that the association between *MER92B_dup23-chr6* and *RNASET2* is unlikely to be driven by LD or shared SNPs. This supports the hypothesis that *MER92B_dup23-chr6* may directly regulate *RNASET2*.

[editorial note: figure redacted]

Three possible explanations for an observed association between a trait and gene expression through genotypes. This figure is from Zhu et al. 2016.

Additionally, epigenetic data (H3K27ac, H3K4me3 ChIP-seq, and ATAC-seq data of CD4⁺ T cells from ENCODE and GeneHancer) suggest that the relationship between *MER92B_dup23-chr6* and *RNASET2* is more likely to reflect a regulatory mechanism rather than mere co-expression.

Fig. 6e Chromatin conformation data of *RNASET2* and *MER92B_dup23-chr6* with significant pleiotropic association. H3K27ac, H3K4me3 ChIP-seq, and ATAC-seq data for CD4-T cells were downloaded from ENCODE⁵. The x axis denotes the physical position along a segment of chromosome 19 containing the *RNASET2* gene and *MER92B_dup23-chr6*. *RNASET2*-interaction genome regions are derived from GeneHancer⁶, which consists of clustered interactions of GeneHancer regulatory elements and genes. ENCODE cCREs represents the candidate *cis*-Regulatory Elements derived from ENCODE⁵. The exon structure of *RNASET2* is presented in the bottom horizontal track.

As stated in the Zhu et al. 2016 paper, this model (SMR and HEIDI) cannot distinguish between causal and pleiotropic associations, that is, it cannot completely rule out the possibility of co-expression. Therefore, we have weakened the article's statement and revised the Discussion section to clarify the limitations of our findings:

The revised text now reads:

“While the SMR analysis provides valuable insights into the potential regulatory relationships between HERVs and autoimmune disease-associated genes, it is important to note that SMR cannot definitively establish causal associations between two phenotypes. Instead, SMR identifies pleiotropic associations, which may arise from shared causal variants.

Despite this limitation, the SMR results offer important clues that can guide further experimental validation. For example, the identification of HERVs significantly associated with autoimmune disease traits highlights potential candidate loci for functional studies. Future experiments, such as CRISPR/Cas9-mediated gene editing, chromatin conformation capture (3C), or reporter assays, could be employed to directly test whether HERVs regulate the expression of nearby genes or influence immune-related pathways. Additionally, integrating multi-omics data (e.g., chromatin accessibility, histone modifications, and transcription factor binding) could provide further mechanistic insights into the regulatory roles of HERVs in autoimmune diseases.”

We greatly appreciate the reviewer’s attention to this important issue. Their feedback has helped us strengthen the clarity and rigor of our manuscript.

Minor points:

How do the identified cis-eQTLs in this manuscript compare to (She et al. 2022)? The authors should comment on how this work compares to HERV eQTL analysis from bulk sequencing data.

Reply: We thank the reviewer for raising this important point. Our study significantly expands upon the HERV eQTL analysis performed by She et al. (2022) in several key aspects:

(1) Sample size: Our analysis is based on single-cell RNA sequencing data from PBMCs of 981 healthy donors, compared to the 315 blood samples analyzed in She et al. (2022). This larger sample size enabled us to identify 2,478 eHERVs, a substantial increase over the 263 eHERVs reported in their study.

(2) Cell type resolution: While She et al. (2022) identified cis-eQTLs at the bulk tissue level, our study provides cell type-specific cis-eQTLs, offering a more granular understanding of HERV regulation across different immune cell types.

These advancements highlight the added value of our work in uncovering the regulatory landscape of HERVs at a higher resolution and with greater statistical power. We have incorporated these comparisons into the revised manuscript to provide a more comprehensive understanding of the regulatory mechanisms of HERVs in immune cells. The revised text now reads:

“Our study significantly expands upon the HERV eQTL analysis performed by She et al⁷ in several key aspects. First, our analysis is based on single-cell RNA sequencing data from PBMCs of 981 healthy donors, compared to 315 blood samples in She et al. This larger sample size enabled us to identify 2,478 eHERVs, a substantial increase over the 263 eHERVs reported in their study. Second, while She et al identified cis-eQTLs at the bulk tissue level, our study provides cell type-specific cis-eQTLs, providing higher resolution insights of HERV regulation across different immune cell types. These advancements highlight the added value of our work in uncovering the regulatory landscape of HERVs at a higher resolution and with greater statistical power.”

In line 102, can the authors clarify the use of significant reads in the following statement “Additionally, 24547 (43%) expressed HERVs were found in intergenic regions (Fig. 1e), supported by significant sequencing reads, indicating programmed expression rather than sequencing noise (Fig. 1f).”

Reply: We thank the reviewer for their question regarding the use of “significant sequencing reads” in line 102. To clarify, the term “significant sequencing reads” refers to the high abundance of sequencing reads mapping to HERV loci in intergenic regions, as visualized in the upper panel of Fig. 1f (now Figure 2c in the revised manuscript). Specifically, the upper panel of Fig. 1f shows a prominent peak in the expression track of a representative HERV (*MLT1F1_dup14-chr9*) located in an intergenic region, indicating robust and reproducible expression rather than random sequencing noise.

Fig 2c Distribution of RNA-seq peaks in the *KLHL9* and *NFATC2IP* locus. *MLT1F1_dup14-chr9* is located in the intergenic region, and *MER21B_dup25-chr16* is located in the antisense strand of the *NFATC2IP* intron.

We have revised the text to make this clearer: “Additionally, 24,547 (43%) expressed HERVs were found in intergenic regions (Fig. 2c). **For example, *MLT1F1_dup14_chr9*, situated in an intergenic region near *KLHL9*, exhibited a prominent read peak on the RNA expression track, suggesting programmed expression rather than sequencing noise.**”

In Figure 2c, the data is re-clustered using highly variable HERVs, are the cells of other cell types that are no longer reflected in the data removed from the analysis or encompassed by the remaining clusters? If the latter, this should be noted that the clusters are not necessarily reflective of the cell types.

Reply: We thank the reviewer for their insightful question regarding Figure 2c (now Figure 3c in the revised manuscript). To clarify, before performing the dimensionality reduction and UMAP visualization analysis, we **removed cells of other cell types (HSC, Plasma, Mega, and gdT)** from the dataset, retaining only the five major immune cell types with sufficient cell counts for further analysis. Additionally, the cell labels in the figure are derived from annotations based on highly variable genes.

In figures 2c and 2e, the UMAP should be displayed with the individual points visible so that it is possible to infer the sample size.

Reply: Thank you for your suggestion regarding Figures 2c and 2e (now Figure 3c and 3e in the revised manuscript). We have updated the UMAP plots to ensure that individual points are visible, allowing for a clearer inference of the sample size.

Fig. 3c UMAP projection based on the expression of highly variable HERVs. Each spot represents a pseudobulk cell type sample of an individual.

Fig. 3e UMAP projection (left) and RNA-seq track (right) of cell type-specific HERV expression.

In figure 2i, the tracks should be extended to include nearby genes that might also be co-regulated.

Reply: Thank you for your valuable suggestion regarding Figure 2i (now Figure 3i in the revised manuscript). As shown below, we have extended the tracks in the figure to include nearby genes.

Supplementary figure 2f is called out in the text but does not exist.

Reply: We thank the reviewer for pointing out this error.

The revised text now reads:

“The cell type-specific HERVs were also enriched in the active chromatin regions of corresponding cell types (Supplementary Fig. 4d-4e).”

In the abstract, the statement is made that 56668 expressed HERV loci with cell type-specific expression were identified. I do not believe the claim is made in the main text that these loci are all cell-type specific.

Reply: We thank the reviewer for pointing out this inaccuracy. The correct number of cell type-specific HERVs is 4532, not 56668. We have revised the text to clearly distinguish between the total number of expressed HERV loci (56668) and those with cell type-specific expression patterns (4532).

The updated text now reads: “Utilizing single-cell RNA sequencing data of peripheral blood mononuclear cells (PBMCs), **we identified 56668 expressed HERV loci, of which 4532 exhibited cell type-specific expression patterns.** These cell type-specific HERVs were predominantly enriched in active genomic regions such as promoters and enhancers.”

Reviewer #3 (Remarks to the Author):

Zhu et al provide an impressive bioinformatic characterization of HERV element expression and activity in the PBMCs of individuals and leverage this data to associate with autoimmune diseases. The data are clean, convincing, and well organized. However, there are instances in the text and referencing in which the authors fail to address certain nuances of HERV elements, which can be admittedly difficult. Otherwise, the authors have approached their results without overreaching and sensationalizing, which is of high importance regarding the work conducted. Overall, the manuscript is impactful, timely, and of high-quality. The writing of the text would benefit from some English editing.

Comments

Line 18: It should be made clear that HERVs are not TEs, and that this is a common misconception and term that is frequently, and inappropriately, utilized by the field in the literature. More accurately, they should instead be referred to and classified under “endogenous retroelements” due to their lack of transposition. Please see the following review in example and for clarification DOI: [10.1038/s41580-023-00674-z](https://doi.org/10.1038/s41580-023-00674-z)

Reply: We sincerely appreciate the reviewer’s insightful feedback regarding the terminology used to classify human endogenous retroviruses (HERVs). We agree that the original statement in Line 18 inaccurately categorized HERVs as “transposable elements (TEs)” and recognize the need for precise terminology in this context.

In response to the reviewer’s suggestion, we have revised the sentence to accurately describe HERVs as “endogenous retroelements” rather than “transposable elements.”

The revised text now reads: “Human endogenous retroviruses (HERVs), **a class of endogenous retroelements**, constitute a significant portion of the human genome and play complex roles in gene regulation and disease processes.”

Line 39: The structure described would be representative of an intact provirus, which is not the case for the majority of HERV elements. Some more descriptive background on the typical structures of modern ERV elements in the human genome would provide a more descript and accurate introduction.

Reply: We thank the reviewer for their valuable feedback. We agree that the described

structure (5'LTR-gag-pro-pol-env-3'LTR) represents an intact provirus, which is not typical for the majority of HERV elements in the human genome. To provide a more accurate and descriptive introduction, we have revised the text to include additional background on the typical structures of modern HERV elements.

The revised text now reads: “Structurally similar to exogenous retroviruses, **intact HERV proviruses** are characterized by the sequence HERVs are characterized by the sequence: 5'LTR-gag-pro-pol-env-3'LTR. **However, due to accumulated mutations and deletions over evolutionary time, the majority of HERV elements in the human genome are incomplete, often lacking one or more of these essential genes.** Among the various forms of HERVs, solo LTRs (long terminal repeats) are the most prevalent, representing a significant fraction of these genomic elements.”

Line 44: It would be ingenuine to relay that HERVs “evolved” to provide utility, it is more so that the relationship with HERVs and the host have evolved to coopt or utilize their sequences. Please rephrase.

Reply: We thank the reviewer for emphasizing the need to avoid implying active evolution of HERVs. We have revised Line 44 to remove "evolution" while retaining the core concept of host-driven co-option:

The updated text now reads: “However, some HERV loci have been **co-opted by the host genome** to serve indispensable roles.”

Line 47: This should read “Solo LTRs are the most common form”

Reply: We thank the reviewer for highlighting this grammatical inconsistency. We have revised Line 47 in the manuscript to correct the subject-verb agreement, as suggested.

The revised text now reads: "Solo LTRs **are** the most common HERV form."

Line 47, 50, 51, and throughout: Please standardize the usage of acronyms at their first occurrence (e.g., LTR and IFN, which is later defined at line 51).

Reply: Thank you for highlighting the inconsistency in acronym usage. We have carefully revised the manuscript to ensure all acronyms are explicitly defined at their first occurrence, and standardized this practice throughout the text. Key revisions include:

Line 47 (Solo LTR):

Added full term at first mention:

"...solo LTRs (**long terminal repeats**) are the most common HERV form..."

Line 51 (IFN):

"For example, HERVs have been shown to act as enhancers of IFN (**interferon**) genes, thereby contributing to the development of transcriptional networks underlying the IFN response, a major component of innate immunity"

Other instances:

Line 26 (eQTL):

"We further identified 4848 conditionally independent *cis*-eQTLs (**expression quantitative trait loci**), associated with HERVs, highlighting their potential role in mediating genetic variants and disease associations."

Line 54: I may be unaware of some literature, but I am unaware of any evidence showing that the progenitor viruses responsible for endogenization events causes any disease other than retroviremia. While intuitive that their would be comorbidities and such, I do not believe there is any evidence of which. Please rephrase or add citations.

Reply: We thank the reviewer for raising this important point. We agree that the comparison to ancestral viral pathogenicity lacks direct evidence and may introduce unintended implications. We have revised Line 54 by **removing the phrase "similarly as their ancestors"** to focus strictly on the documented roles of reactivated HERVs in human diseases.

The revised text now reads: "Despite these programmed incorporations of HERV elements in normal cellular processes, some HERV loci may become reactivated, contributing to human diseases such as tumor and autoimmune diseases."

Line 64: other than these 2 examples, there are pretty substantial advancements that have been main regarding HERV activity and immunity that also warrant discussion. Please expand upon this section of the introduction. Furthermore, HERV RTase activity is quite controversial within the field and some of the better described mechanisms (e.g., transcriptional regulation and immunogenic RNA production) warrant discussion as well. DOI: 10.1146/annurev-immunol-101721-033341

Reply: We thank the reviewer for their valuable feedback. We agree that the discussion

of HERV activity and immunity in the introduction should be expanded to include additional advancements and mechanisms. Following the reviewer's suggestion and the referenced review (DOI: 10.1146/annurev-immunol-101721-033341), we have revised the text to incorporate more comprehensive information on HERV interactions with the immune system, including transcriptional regulation and immunogenic RNA production. Additionally, we have cited the recommended review (DOI: 10.1038/s41580-023-00674-z) in the revised manuscript.

The revised text now reads: Our immune system is essential for protecting us from exogenous viral infections, and studies have shown that HERVs can also interact with the immune system. For example, the reverse transcription products of HERVs may induce aging and inflammatory responses by activating the cGAS-STING innate immune pathway⁸. **Additionally, HERV-derived RNAs can act as immunogenic molecules, triggering pattern recognition receptors (PRRs) such as Toll-like receptors (TLRs) and RIG-I-like receptors (RLRs), thereby eliciting antiviral immune responses. Furthermore, HERVs can contribute to immune dysregulation by producing viral-like proteins or peptides that mimic self-antigens, potentially leading to autoimmune responses⁹.** Researchers have also observed a higher HERV expression in the PBMCs¹⁰ of aged individuals, suggesting a potential link between HERV activity and immunosenescence.

Line 91: I agree with this methodology, please just indicate why HERVs overlapping with gene exons were excluded for any readers who are unfamiliar with why this was done.

Reply: Thank you for suggesting this clarification. We have added a brief explanation to address why HERVs overlapping gene exons were excluded.

The revised text now reads: "To systematically assess HERV expression across these cell types, we constructed a comprehensive HERV annotation GTF file using the UCSC Table Browser (<https://genome.ucsc.edu/cgi-bin/hgTables>). **We excluded HERV loci overlapping gene exons to minimize confounding effects from host gene transcription signals and to focus on HERVs with potential independent regulatory roles.**"

Figure 1D: I am always quite interested in these sorts of plots and would appreciate a more in-depth demonstration of this by individual and by cell type. It would be

incredibly informative to add an additional supplemental figure, which can also be referred to on line 97, that breaks down these gross expression profiles as such, and include statistical testing when appropriate. This bulked analysis is fine for the main text, but how does this differ amongst cell types across individuals?

Reply: We sincerely thank the reviewer for their insightful comment regarding the expression profiles of HERVs across individuals and cell types. Here, we provide a detailed response to address this concern:

In response to the reviewer’s suggestion, we have conducted a more in-depth analysis of HERV expression profiles by individual and cell type. Specifically, we analyzed the proportion of each HERV family in individual cell types (**Supplementary Fig. 2a**) and their distribution across age groups within each cell type (**Supplementary Fig. 2b**). Notably, our analysis revealed no significant differences in HERV family proportions across age groups within any cell type, suggesting that the expression of HERV families is relatively stable over time within specific immune cell populations.

To incorporate these findings, we have revised the text as follows:

“We examined the expressed HERV families and found that ERV1 and ERVK families were more active in PBMCs (Fig. 2b), possibly due to their relatively recent integration into the human genome. **To further investigate the expression patterns of HERV families across different cell types and age groups, we analyzed the proportion of each HERV family in individual cell types (Supplementary Fig. 2a) and their distribution across age groups within each cell type (Supplementary Fig. 2b). Notably, our analysis revealed no significant differences in HERV family proportions across age groups within any cell type, suggesting that the expression of HERV families is relatively stable over time within specific immune cell populations.**”

Supplementary Fig. 2a Distribution of HERV families across five cell types.

Supplementary Fig. 2b Temporal dynamics of HERV family proportions across each cell type. The p-value was derived by calculating the Pearson correlation coefficient.

Figure 2C: I'm impressed by how effective and clean this UMAP projection came out. Can you add a breakdown of how many of these HERVs are intronic vs intergenic?

Reply: We sincerely thank the reviewer for their positive feedback on the UMAP projection in Figure 2c and for their insightful suggestion to include a breakdown of intronic vs intergenic HERVs.

In response to the reviewer's comment, we have updated Figure 2b (now Figure 3b in the revised manuscript) to include the distribution of HERVs across different genomic regions. Specifically, we identified 1264 HERVs in distal intergenic regions, 956 in introns on the same strand, 804 in introns on the opposite strand, 167 in promoter regions, 4 in downstream regions, and 2 in 3' UTR regions.

Fig. 3b Left panel: scatter plot of highly variable HERVs (dispersion > 0.5, expression > 0.1). Right panel: scatter plot of the spearman's correlation coefficient of highly variable HERVs with related genes.

Line 218: Can you clarify what you mean by this statement regarding an interaction with 7 HLA genes? Similarly, BRD2.

Reply: We thank the reviewer for this question regarding the interaction with HLA genes. In the manuscript, we stated that *MLT10_dup50-chr6* interacts with 7 HLA genes based on data from GeneHancer. Here is a detailed clarification: GeneHancer is a comprehensive database that integrates enhancer-gene interactions from multiple sources. It predicts regulatory interactions between genomic elements (e.g., enhancers) and genes. For *MLT10_dup50-chr6*, GeneHancer records interactions with 7 HLA genes and 4 other genes (e.g., *BRD2*), all located in the major histocompatibility complex (MHC) region on chromosome 6. These interactions suggest that *MLT10_dup50-chr6* may regulate or influence the expression of these genes. We hope this clarifies the statement.

Line 223: This would more be attributed to putative roles of HERVs to impact B cell functionality to contribute to autoinflammation more so than the roles of B cells in autoimmunity. Please rephrase.

Reply: We thank the reviewer for this insightful comment. We agree that the observed enrichment of autoimmune disease-associated HERVs in B cells may be more closely linked to the potential roles of HERVs in modulating B cell functionality and contributing to autoinflammation.

To address this point, we have revised the relevant sentence in the manuscript as follows: **“This may be attributed to the potential roles of HERVs in modulating B cell**

functionality and contributing to autoinflammation.”

Line 242: While the section itself is fine in the language, the heading needs to be rephrased. While this is impressive bioinformatic work, without wet lab validation it would be ingenious. Please soften to “HERVs potentially regulate autoimmune disease-associated genes” or something of that manner.

Reply: We thank the reviewer for their constructive feedback on the section heading. We agree that the original heading may have overstated the conclusions without wet lab validation. To better reflect the preliminary nature of our findings and to align with the reviewer’s suggestion, we have revised the heading to a more cautious and accurate phrasing.

The revised heading now reads: “HERVs **Potentially** Regulate Autoimmune Disease-Associated Genes”

Line 291: This sentence does not make much sense and is attempting to combine two separate thoughts. Please divide this into two statements that are better relayed.

Reply: We thank the reviewer for pointing out the lack of clarity in this sentence. We agree that the original sentence attempted to combine two distinct ideas, which may have caused confusion. To address this, we have divided the sentence into two separate statements for better clarity and flow.

The revised text now reads: “**Studies have shown that HERVs are involved in various biological processes and complex diseases. These findings largely rely on observed differences in HERV family expression.**”

Line 293: Numerous studies and bioinformatic pipelines have permitted the determination of locus-specific HERV expression from numerous datasets. While issues from multimapping, dropouts, read length, and number of reads from certain NGS datasets make this difficult, the problem has been approached and accomplished numerous times. This statement is both dated and no longer factual. Delete entirely.

Reply: We thank the reviewer for their feedback. In accordance with your suggestion, we have removed the statement regarding locus-specific HERV expression determination from the manuscript.

Line 304: The way this statement reads makes it seem as if the authors are the first to apply similar methodologies to perform locus-specific quantification of HERV elements from short read scRNA sequencing data, which is simply not the case. Please revise the last statement of this paragraph to better relay that.

Reply: We thank the reviewer for pointing out this issue. We agree that the original statement may have inadvertently implied that we were the first to apply locus-specific quantification of HERV elements from short-read scRNA-seq data, which was not our intention. To clarify, we have revised the sentence to better reflect the existing literature and the potential of our approach.

The revised text now reads: “Nevertheless, this locus-specific approach, **building on established methodologies**⁴, could be extended to studies of other cell types, further advancing our understanding of the functions of HERV elements in the human genome.”

Line 306: Similar to my first comment regarding line 18, please ensure that the most correct nomenclature is utilized.

Reply: We thank the reviewer for pointing out this important distinction. We agree that HERVs should not be classified as transposable elements (TEs) due to their lack of active transposition. Following the reviewer’s suggestion and the referenced review (DOI: 10.1038/s41580-023-00674-z), we have revised the sentence to more accurately describe HERVs as endogenous retroelements.

The revised text now reads: “HERVs are categorized as **endogenous retroelements** within the human genome, reflecting their origin from ancient retroviral infections and their lack of active transposition.”

Line 315: I am confused by what the authors mean in “rather than typical gene expression pathways”, as if epigenetics are not part of typical genetic regulation? Please revise this statement to be clearer by what is mean.

Reply: We thank the reviewer for pointing out the ambiguity in this statement. We agree that epigenetic mechanisms are indeed part of typical gene regulation. To clarify, we remove the statement of “rather than typical gene expression pathways”. The revised text now reads:

“These findings suggest that HERV expression is primarily regulated by epigenetic mechanisms.”

Line 328: Please delete the sentence beginning on this line. It is not necessary and would likely be a source of contention across multiple camps of people in retrovirology, paleovirology, genetics, etc.

Reply: We thank the reviewer for their feedback. In accordance with your suggestion, we have removed this sentence.

Line 339 and potentially elsewhere: I think it is important to change the language from “.. ∙∙ they can facilitate ∙∙ ..” to something along the lines of “.. ∙∙ their deregulated activity may facilitate ∙∙ ..”. Importantly, the presence of HERV elements alone is not a nefarious detriment to human health, and conditions need to be met for their deleterious effects. This may also require better defining elsewhere in the document as well.

Reply: We thank the reviewer for their valuable feedback. We agree that the language regarding the potential effects of HERVs should be more precise to reflect that their deleterious effects are conditional rather than inherent. Following the reviewer’s suggestion, we have revised the relevant sentence to better convey this nuance.

The revised text now reads: “Overall, HERVs in the human genome exhibit multifaceted roles: on one hand, they serve as indispensable elements and genes in human evolution; on the other hand, **when their expression is deregulated, they may contribute** to abnormal processes such as aging, tumor development, and autoimmune diseases.”

Discussion: Due to the bioinformatic nature of the article, it is incredibly important that the authors provide a paragraph on the limitations of the study. This should be added as the second to last paragraph.

Reply: We thank the reviewer for their valuable suggestion. In response, we have added a dedicated paragraph in the Discussion section (second-to-last paragraph) to address the limitations of our study, particularly the lack of experimental validation.

The new paragraph is as follows:

“While the SMR analysis provides valuable insights into the potential regulatory relationships between HERVs and autoimmune disease-associated genes, it is important to note that SMR analysis cannot definitively establish causal associations between two phenotypes. Instead, SMR identifies pleiotropic associations, which may arise from shared causal variants. In our study, the observed associations between HERVs and autoimmune disease traits could reflect either direct regulatory effects of HERVs or indirect effects mediated by nearby genes or other genomic elements.

Despite this limitation, the SMR results offer important clues that can guide further experimental validation. For example, the identification of HERVs significantly associated with autoimmune disease traits highlights potential candidate loci for functional studies. Future experiments, such as CRISPR/Cas9-mediated gene editing, chromatin conformation capture (3C), or reporter assays, could be employed to directly test whether HERVs regulate the expression of nearby genes or influence immune-related pathways. Additionally, integrating multi-omics data (e.g., chromatin accessibility, histone modifications, and transcription factor binding) could provide further mechanistic insights into the regulatory roles of HERVs in autoimmune diseases.”

Reviewer #3 (Remarks on code availability):

Briefly reviewing and I am not a bioinformatician, but the code appears well organized to my naive eyes.

1. He, J. *et al.* Identifying transposable element expression dynamics and heterogeneity during development at the single-cell level with a processing pipeline scTE. *Nat Commun* **12**, 1456 (2021).
2. Rodríguez-Quiroz, R. & Valdebenito-Maturana, B. SoloTE for improved analysis of transposable elements in single-cell RNA-Seq data using locus-specific expression. *Commun Biol* **5**, 1063 (2022).
3. Berrens, R.V. *et al.* Locus-specific expression of transposable elements in single cells with CELLO-seq. *Nature Biotechnology* **40**, 546-554 (2022).
4. Wang, J. *et al.* Single-cell RNA sequencing highlights the functional role of human endogenous retroviruses in gallbladder cancer. *EBioMedicine* **85**, 104319 (2022).
5. Consortium, E.P. An integrated encyclopedia of DNA elements in the human genome. *Nature* **489(7414)**, 57-74 (2012).

6. Fishilevich, S. *et al.* GeneHancer: genome-wide integration of enhancers and target genes in GeneCards. *Database (Oxford)* **2017**(2017).
7. She, J. *et al.* The landscape of hervRNAs transcribed from human endogenous retroviruses across human body sites. *Genome Biol* **23**, 231 (2022).
8. Liu, X. *et al.* Resurrection of endogenous retroviruses during aging reinforces senescence. *Cell* **186**, 287-304 e26 (2023).
9. Bonaventura, P. *et al.* Identification of shared tumor epitopes from endogenous retroviruses inducing high-avidity cytotoxic T cells for cancer immunotherapy. *Science Advances* **8**, eabj3671 (2022).
10. Autio, A. *et al.* Effect of aging on the transcriptomic changes associated with the expression of the HERV-K (HML-2) provirus at 1q22. *Immunity & Ageing* **17**, 11 (2020).

Reviewer #1 (Remarks to the Author)

The authors have addressed most of my concerns, and I remain enthusiastic about the study. However, one major issue remains that should be addressed before publication.

The current pipeline quantifies HERV expression by aligning the same reads separately to two references: one for human genes and one for HERVs. This approach introduces a key flaw: reads that appear uniquely mapped in one reference may actually map equally well to loci in the other, but this ambiguity is hidden by performing alignments independently. As a result, HERV expression may be overestimated due to misclassified multi-mapping reads.

This problem is especially relevant for repetitive or homologous sequences like HERVs, and is analogous to known issues in mitochondrial RNA analysis, where reads from NUMTs can be misassigned unless aligned to a combined genome index.

By contrast, tools like CELLO-seq, scTE, and soloTE use a single, comprehensive reference for alignment, followed by post-processing to assign reads. This ensures that mapping ambiguity is properly handled.

To improve mapping accuracy and avoid inflation of HERV expression, the authors should revise their pipeline to align reads once to a combined reference containing both genes and HERVs, and then apply appropriate read attribution.

Reply: We sincerely appreciate your insightful feedback regarding mapping specificity, which has greatly strengthened our study. Following your recommendation, we have implemented the following revisions:

1. Pipeline Revision:

We created a combined reference GTF file incorporating both gene and HERV annotations. All samples were reprocessed using Cell Ranger (v7.1) with this unified reference. The alignment script used was:

```
cellranger count \  
--fastqs "fastqs" \  
--sample = "sample_name" \  

```

```
--localcores 16 \  
--id = "new_matrix" \  
--transcriptome = "/genome/combined_HERV_gene_reference/" \  
--include-introns="False".
```

Only uniquely mapped reads were retained for gene and HERV expression quantification. We have updated the **Methods** and **Results** section to reflect these improvements. The revised text now reads:

“Construction of HERV-Gene Integrated Reference and Expression Quantification

The HERV annotation file, compiled by RepeatMasker, was obtained from the UCSC Table Browser for GRCh38 (<https://genome.ucsc.edu/cgi-bin/hgTables>). HERVs that partially overlapped with gene exons were removed to avoid quantitative bias. As a result, 692759 HERVs sites were obtained to provide annotation information for the subsequent quantitative process (Supplementary Table 1). Using CellRanger (v7.1.0), we then constructed a combined reference genome incorporating both filtered HERV annotations and protein-coding genes through the “mkref” function.

PBMC Single-cell RNA-seq data (fastq file) were downloaded from GSE196830¹⁶ and syn50209110²⁶. Data processing used CellRanger (v7.1.0) “count” function with these parameters: --fastqs “./fastqs” --sample = “sample” --localcores 16 --id = “new_matrix” --transcriptome = “/genome/combined_HERV_gene_reference/” --include-introns=“False”. This integrated approach enabled simultaneous profiling of host gene expression and retroelement activity while minimizing cross-mapping artifacts.”

The merged reference GTF file is also updated on GitHub (https://github.com/YoungLi88/HERV_eQTL) to ensure reproducibility and facilitate future research.

2. Updated results:

The revised pipeline identified 41,460 robust HERVs after filtering ambiguous mappings. All figures (Fig. 1–6, S1–S8), supplementary tables (Table S1–S8), and manuscript text have been updated using the new expression matrices. Importantly, our core findings remain well supported.

We are grateful for this suggestion that has enhanced both the technical rigor and biological validity of our conclusions.

Reviewer #1 (Remarks on code availability):

Did not run the code, but pleased that it looks comprehensive.

Reply: We appreciate the reviewer's positive acknowledgment of our code repository.

We welcome any future questions about implementation or usage.

Reviewer #2 (Remarks to the Author):

The responses and revisions made by the authors have addressed my questions and concerns. I suggest that in the methods or text the authors further clarify that the data was subsetted for visualization and analysis in figure 3c prior to re-clustering.

Reply: Thank you for your valuable suggestion. We have carefully revised the Methods section to clarify our analytical workflow as recommended.

The updated text now reads:

"Identification of Highly Variable HERVs and UMAP Visualization

Highly variable HERVs (n=2,045) were identified using the “sc.pp.highly_variable_genes()” function in Scanpy (v1.9.8) with parameters: min_disp=0.5, min_mean=0.1, and max_mean=4.

Prior to UMAP visualization, we subset the dataset by removing rare cell populations (HSCs, plasma cells, Megakaryocytes, and $\gamma\delta$ T cells), retaining only the five major immune cell types (CD4-T, CD8-T, NK, B, and Myeloid cells) with sufficient cell counts. UMAP dimensionality reduction were using “sc.tl.umap()” function in Scanpy (v1.9.8) with parameters: n_neighbors=40, n_pcs=5, res=0.5. Additionally, the cell labels in the figure are derived from annotations based on highly variable genes.”

We appreciate your insightful comment, which has helped improve the methodological transparency of our study.

Reviewer #3 (Remarks to the Author):

The authors did a great job at improving the article which is of timely importance and interest. Some minor comments remain.

Line 52 – Remove "flanking". Just LTRs would be fine and it reads like they must flank genes from both ends to impact expression.

Reply: Thank you for this helpful suggestion. The revised text now reads:

“Studies have shown that **LTRs** can function as promoters or enhancers, regulating the expression of adjacent genes.”

Line 58 – the sentences ending and beginning here could be worded more clearly. Consider rephrasing to “where that may contribute to tumorigenesis^{7,8} autoimmunity^{9,10}, and senescence^{11,12}.”

Reply: Thank you for this helpful suggestion. The revised text now reads:

“Despite these programmed incorporations of HERV elements in normal cellular processes, some HERV loci may become reactivated, **where that may contribute to tumorigenesis^{7,8} autoimmunity^{9,10}, senescence¹¹, and aging¹².**”

Figure 1 – what were the proportions of reads denoted as multimapping using the parameters described? I worry this may be a substantial portion of potentially relevant information. I do believe the methodology is stringent and sound as possible. Other approaches typically apply EM or something to try to compensate for this, albeit also not perfectly or with as many or more flaws inherently. It would be good to record in a supplementary table or something the number/proportion of discarded reads per sample in order to convey what could be deemed as missing information to the reader. Apologies is this is already done and I missed it.

Reply: To improve mapping specificity and prevent HERV expression inflation, we constructed a merged genome reference integrating both protein-coding genes and HERV annotations. Using Cell Ranger (v7.1), we realigned all sequencing data to this unified reference. We assessed the proportion of unique reads for each HERV and found

that most HERVs exhibited a proportion close to 100% (**Supplementary Fig. 1a**), indicating that HERV expression was predominantly driven by unique reads.

Supplementary Fig. 1a The kernel density plot illustrates the distribution of the percentage of unique reads mapped to HERVs.

Line 116 – In regards to SoloTE, I’m not sure if random assignment is the best phrase to characterize EM-based approaches. Inferred assignment would be much better and sound less biased.

Reply: We sincerely appreciate this insightful technical suggestion. After carefully reviewing the SoloTE publication (<https://www.nature.com/articles/s42003-022-04020-5#Sec2>), we confirm that their current implementation indeed employs random assignment of multi-mapping reads, as clearly illustrated in Fig. 1a legend: “SoloTE assigns reads to each locus, with multi-mapped reads randomly allocated to one location.” As the reviewer astutely notes, the authors propose in their Discussion that “Further improvements to our pipeline are the integration of the Expectation–Maximization (EM) algorithm”, acknowledging the potential benefits of probabilistic methods.

Line 408 – “HERVs in the human genome” reads a bit redundant. Just phrase it as “HERVs” or “ERV in the human genome”

Reply: We sincerely appreciate this constructive suggestion to enhance textual precision. In accordance with your recommendation, we have revised the phrasing from “HERVs in the human genome” to “HERVs” throughout the manuscript.

The modified text now reads:

“Overall, **HERVs** exhibit multifaceted roles...”